# Distinguishing preferences of human APOBEC3A and APOBEC3B for cytosines in hairpin loops, and reflection of these preferences in APOBEC-signature cancer genome mutations

Yasha Butt[1,5], Ramin Sakhtemani [1,2,3,5], Rukshana Mohamad-Ramshan[1], Michael S. Lawrence [2,3] & Ashok S. Bhagwat [1,4] ✉

The APOBEC3 enzymes convert cytosines in single-stranded DNA to uracils to protect against viruses and retrotransposons but can contribute to mutations that diversify tumors. To understand the mechanism of mutagenesis, we map the uracils resulting from expression of APOBEC3B or its catalytic carboxy-terminal domain (CTD) in *Escherichia coli*. Like APOBEC3A, the uracilomes of A3B and A3B-CTD show a preference to deaminate cytosines near transcription start sites and the lagging-strand replication templates and in hairpin loops. Both biochemical activities of the enzymes and genomic uracil distribution show that A3A prefers 3 nt loops the best, while A3B prefers 4 nt loops. Reanalysis of hairpin loop mutations in human tumors finds intrinsic characteristics of both the enzymes, with a much stronger contribution from A3A. We apply Hairpin Signatures 1 and 2, which define A3A and A3B preferences respectively and are orthogonal to published methods, to evaluate their contribution to human tumor mutations.

The apolipoprotein B mRNA-editing catalytic polypeptide-like subfamily 3 (APOBEC3) enzymes are a part of the human innate immune response against viruses and retrotransposons[1–9]. They belong to a family of DNA-cytosine deaminases that convert cytosines to uracils. Error-prone repair or replication of these uracils creates predominantly C:G to T:A or G:C mutations, within specific trinucleotide contexts (generally TCN, where N is any nucleotide) and are referred to as APOBEC signature mutations[10–13]. While these enzymes function to restrict viral growth and retrotransposition by targeting single-stranded DNA (ssDNA) intermediates in these processes, they can also cause mutations in the human genome. Several different types of human tumors contain APOBEC signature mutations, and they aid

evolution of tumor genomes and help the tumors avoid anticancer therapy[14–17]. In particular, two members of this subfamily, APOBEC3A (A3A) and APOBEC3B (A3B) have been implicated in causing tumor mutations[18–20]. Consequently, elucidating their preferences for nucleotide sequences and nucleic acid secondary structure is of vital importance in understanding their biological roles.

A3A and A3B, which are likely to have formed from a gene duplication event[21,22], have diverged in important ways. While A3A contains a single zinc-binding domain, A3B contains two such domains[23]. Of the two domains within A3B, the carboxy-terminal domain (CTD) domain shares an 89% sequence identity with A3A and is catalytically active[3,23] whereas the amino-terminal domain (NTD) of

---

[1]Department of Chemistry, Wayne State University, Detroit, MI 48202, USA. [2]Massachusetts General Hospital Cancer Center, Boston, MA, USA. [3]Broad Institute of MIT and Harvard, Cambridge, MA, USA. [4]Department of Biochemistry, Microbiology and Immunology, Wayne State University School of Medicine, Detroit, MI 48201, USA. [5]These authors contributed equally: Yasha Butt, Ramin Sakhtemani. ✉e-mail: axb@chem.wayne.edu

A3B is inactive[24,25]. While A3B resides principally in the nucleus[9,26–28], A3A is thought to shuttle between the nucleus and the cytoplasm[29]. The two enzymes also differ in their interactions with RNA. Whereas A3A has been shown to weakly deaminate cytosines in RNA[30], such an activity has not been reported for A3B. Additionally, A3B activity is greatly reduced through its binding to RNA[18].

Despite the structural and biological differences between the two enzymes, there are many similarities between the patterns of mutations caused by them. Both the enzymes prefer a thymine to the 5′ side of the cytosine (5′TC) and preferentially target the lagging-strand template (LGST) within replication forks[31–34]. However, there have been differing claims regarding the mutagenicity of the two enzymes in human cells and to what extent they contribute to mutations in cancer cells. Based in part on its expression in breast cancer-derived cell lines and tumors, one group of investigators concluded that A3B is the source of mutations in breast cancer cells [35] but subsequent studies showed that while A3B is overexpressed in many tumor cells[18,35,36], it is generally bound to RNA, which largely inactivates it[18,37]. Additionally, A3A activity was much less affected by cellular RNA and A3A expression better correlated with occurrence of APOBEC signature mutations than A3B [18].

The mutations in tumor genomes that principally occur at cytosines in TC dinucleotides that do not overlap CG dinucleotides and are either C:G to T:A transitions or C:G to G:C transversions are referred to as APOBEC signature mutations[10,11] and there has been considerable effort in recent years to determine whether A3A or A3B is principally responsible for them. One study showed that A3A and A3B cause mutational clusters in the yeast genome that are characteristic of breast cancer mutations[38] and another study showed that C:G to T:A mutations in a reporter gene in yeast caused by A3A preferentially occur in YTCA sites, whereas A3B favors RTCA sites (Y is a pyrimidine and R is a purine[39]).

A subsequent study of cancer genome mutations found that APOBEC signature mutations were highly enriched in hairpin loops[40,41]. Furthermore, the sequence context of these mutations was more consistent with being caused by A3A, than by A3B[39,42]. Biochemical assays using cell-free extracts containing these enzymes confirmed that A3A has a strong preference for cytosines within TC sequences with the cytosine at the 3′ end of hairpin loops, but A3B had roughly the same activity on many hairpins and linear substrates[40,41]. This led to the conclusion that A3A and not A3B is the main cause of recurrent APOBEC mutations in DNA stem-loops in tumors[40]. A related study[43], showed that the ssDNA-binding protein, RPA, can affect the activity of A3A, generally inhibiting it. This has led to the suggestion that there is a competition between RPA and the APOBECs for binding to ssDNA, especially at replication forks[41,44,45].

We developed an experimental methodology which has addressed these questions using the *E. coli* genome as the target for these enzymes. Following expression of one of the AID/APOBEC family of enzymes in an UNG⁻ strain of *E. coli*, the genomic DNA is extracted and deoxyuridines are converted to abasic sites. The abasic sites are reacted with a chemical containing a detachable biotin. The tagged DNA fragments are pulled down using streptavidin beads, released from the beads and are sequenced and mapped to the bacterial genome[46]. This strategy, which is termed uracil pull-down sequencing (UPD-seq) shows the distribution of uracils in the genome (uracilome) and has been applied to A3A and human activation-induced deaminase (AID). This analysis confirmed that A3A targets hairpin loops[46] and further showed that cytosines in VpC dinucleotides (V is A, C or G) are much better targets for A3A when they appear in hairpin loops than in linear DNA[41]. UPD-seq also showed that AID lacks a strong preference for LGST in replication forks and does not prefer cytosines in hairpins[44]. Normal B lymphocytes undergoing somatic hypermutation and class-switch recombination express AID during the G1 phase of the cell cycle[47], and hence these

results were interpreted to mean that ssDNA in replication forks is effectively protected by RPA from the action of any AID that may leak into the S phase[44].

We report here results of UPD-seq performed on *E. coli* expressing full-length A3B (A3B-full) or A3B-CTD and compare the results with those from similar analysis of A3A expressing cells. Surprisingly, we find that A3B, like A3A also preferentially deaminates cytosines in hairpin loops, but the preferred sequence contexts of the two enzymes are somewhat different.

## Results

### *E. coli* uracilomes of both A3B-full and A3B-CTD are similar to A3A uracilome

The genomic distribution of uracils, the uracilome, created by full-length A3B (A3B-full) and the carboxy-terminal domain of A3B (A3B-CTD) was determined in multiple independent samples using UPD-seq (see Introduction above[46]) and the results were analyzed to determine the types of biochemical, structural and nucleotide sequence features that affect uracil accumulation. The results for the two sets were then compared with each other and with the uracilome of A3A[46]. The peaks of uracilation were determined within all the uracilomes as described previously in ref. 46 (Supplementary Fig. S3A) and overlapping peaks within different uracilomes were identified. This analysis found that all 14 peaks common within the three independent A3B-CTD samples overlapped peaks common within the A3B-full samples (Supplementary Table S1 and Fig. 1A). However, there were 19 peaks common within six A3B-full datasets that were not found in A3B-CTD datasets (Supplementary Table S1), which is consistent with the previous conclusion that the activity of A3B-full in *E. coli* is higher than that of A3B-CTD[25,48]. Despite the presence of greater numbers of peaks within the A3B-full uracilome, both the enzymes appeared to target similar features of the genome. These included a high preference for tRNA genes (10 out of 87 genes for A3B-CTD and 35 genes for A3B-full; Supplementary Table S1), transcription start sites (TSS; 10 out of 15 peaks for A3B-CTD and 28 out of 35 peaks for A3B-full; Supplementary Fig. S4) and the lagging-strand template (LGST; Fig. 1B). The tendency of AID/APOBEC enzymes of preferentially deaminating cytosines near TSS and within tRNA genes in *E. coli*, yeast and human tumors has been noted previously[44,46,49,50].

Seven peaks in the uracilomes of A3B-CTD and A3B-full overlapped with peaks in the A3A uracilome (Fig. 1A and Supplementary Fig. S5). Furthermore, the preferences of A3B for tRNA genes, TSS and LGST were also previously reported for the A3A uracilome [in ref. 46 and Figs. 1A and 1B] suggesting that the strong sequence similarity between A3B-CTD and A3A (89% identity) causes similar interactions with DNA. These results also suggest that the N-terminal domain of A3B does not have a strong influence on the types of genomic regions preferentially targeted by A3B.

### Like A3A, A3B also prefers cytosines in hairpin loops

In contrast to previous work that has suggested that A3A, but not A3B, prefers cytosines in DNA loops over those in extended chains[40], we found that both A3B-CTD and A3B-full strongly prefer cytosines in hairpin loops in the *E. coli* genome (Fig. 2). The normalized uracilation index (UI), which is a measure of the frequency at which specific cytosines are converted to uracils (Supplementary Fig. S3B), was six to nine times higher for cytosines in hairpin loops than those in non-hairpin DNA when analyzing A3A, A3B-CTD and A3B-full uracilomes (Fig. 2A and Supplementary Fig. S6). The UI increased with increasing hairpin stem strength suggesting that cytosines in the loops of stable hairpins were more likely to be targets of these enzymes than those in less stable hairpins or non-hairpins (Fig. 2B and Supplementary Fig. S6). When the different UPD-seq datasets were compared to each other for similarities in targeting hairpin loops, the data sets for each enzyme had highest correlation with themselves (e.g., the four A3A

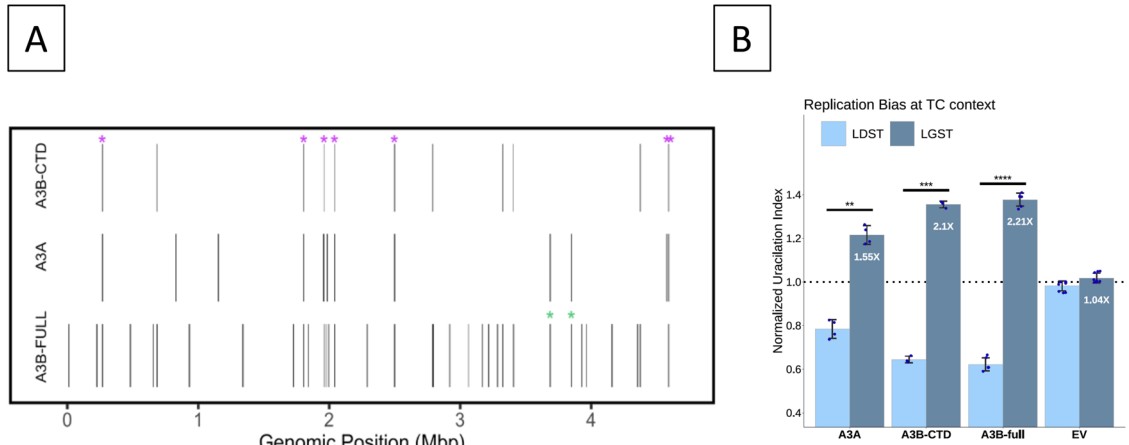

**Fig. 1 | Properties of uracilation by A3B in *E. coli*. A** A bar code plot of uracilation peaks due to A3A, A3B-CTD, and A3B-full in the *E. coli* genome. Each bar represents a region in the genome where all independent samples for an enzyme had a uracilation peak. The peaks marked with a magenta asterisk (*) are common to all three datasets and the green asterisks represent peaks common to A3A and A3B-full, but are not found in the A3B-CTD data. The last magenta asterisk has two overlapping peaks (Supplementary Fig. S2). **B** Replicative strand bias in uracilation. Normalized uracilation of LGST and LDST for each enzyme is shown. The uracilation of each strand is normalized with respect to the mean uracilation of the genome. The number of biologically independent data sets in this and other figures involving UPD-seq of *E. coli* expressing A3A, A3B-CTD and A3B-full are- A3A (n = 4), A3B-CTD (n = 3), A3B-full (n = 6) and EV (n = 9). Bar heights and whiskers respectively represent mean and standard deviation. A paired sample t-test compared the A3A, A3B-CTD, A3B-full or EV values for LDST and LGST DNA strands. The bars with high statistical significance are marked with "*" ($P \le 0.05$), "**" ($P \le 0.01$), "***" ($P \le 0.001$) or "****" ($P \le 0.0001$). The exact *P*-values are: A3A- 0.002041; A3B-CTD- 0.0006439; A3B-full: $6.7 \times 10^{-7}$. Source data are provided as a Source Data file.

datasets with each other) and there was also high correlation between A3B-full and A3B-CTD data (Fig. 2C). Additionally, most A3B sets showed a moderate correlation with all the A3A sets (mean correlation coefficient, *R*, 0.37; Fig. 2C and Supplementary Table S2). These data suggest that A3A and A3B target hairpins with similar, but non-identical, loop sizes and sequences.

A3B-CTD and A3B-full both targeted loops of similar size and sequences and deaminated cytosines at the 3' end of loops of 3, 4 and 5 nucleotides most frequently (Fig. 2D and Supplementary Data 1) and they had similar preferences for loop sequences (Fig. 2E). Cytosine within the loop sequence TGTC was the best target for both enzymes and a plot of UI values for the two enzymes fitted a linear equation with an R-squared of 0.88 (Fig. 2E). Thus, not only do both forms of A3B prefer cytosines in hairpin loops, but the finer details of this preference are qualitatively the same for them.

In contrast, although both A3A and A3B preferentially target hairpin loops, they prefer loops of different sizes and sequences. A3A is prone to deaminate cytosines in three-nucleotide loops more readily than larger loops[46] (Fig. 2D), while both A3B-CTD and A3B-full prefer four nucleotide loops over other loop sizes (Fig. 2D, Supplementary Fig. S6 and Supplementary Data 1). Furthermore, there was low correlation between UI of A3A and A3B for loop sequences of 3, 4, or 5 nt (Fig. 2F). Interestingly, all three enzymes have a preference for cytosines at the 3'-end of the loops for 3 or 4 nt loops, but not 5 nt loops (Fig. 2D). Whereas A3A prefers loops with the cytosine at position 4 within a 5 nt loop, A3B-full and A3B-CTD prefer the 3' end position (i.e., 5th position; Fig. 2D). These results provide additional evidence that the A3B-CTD is the main determinant of substrate selection in A3B, and show that A3A and A3B enzymes have similar but distinguishable, loop size and sequence preferences.

To understand the sequence context of the cytosines deaminated by the three enzymes, we used river plots[51]. We chose river plots instead of the more common sequence logo plots because the former are able to capture relational information between enriched bases (Supplementary Fig. S7). When the sequence context of all cytosines with a UI > 0.04 was analyzed in this fashion, both A3A and A3B showed preference for a thymine at the −1 position (the deaminated cytosine at position 0) and weaker preferences for bases at other positions

(Fig. 3A). The preference for a thymine at −1 became even stronger when only hairpin loops were considered and guanine was the preferred base at +1 for the hairpins (Fig. 3B). When the positional preferences were analyzed for loops of different sizes, significant differences between A3A and A3B emerged at the −2 position upstream of a TC. When all hairpin loops were considered, A3A preferred T > A > C > G, while A3B preferred A > T > G > C (Fig. 3B). In both the global and hairpin cytosine targets A3A showed a C + T > A + G and A3B showed the reverse preference at −2 position (Fig. 3A, B). The magnitude of these preferences vary for different loop sizes and positions, e.g., A3B shows a much weaker preference for A + G at the 3' position in 4 nt loops than in 5 nt loops (Fig. 3C, D). Regardless, the previous observation based on a yeast mutational study that A3A and A3B respectively prefer a pyrimidine and a purine at −2 position when the cytosine is in a TC dinucleotide in ssDNA[39] is also true for both global and hairpin targets in the *E. coli* genome.

**Deamination activity of A3B-CTD on cytosines in hairpin loops**

In a previous report[40] it was noted that several hairpin sequences were poorer substrates for A3B than A3A. In particular, a hairpin with GTT.C. (the cytosine flanked by dots is deaminated) as the loop sequence was only as good a substrate as its linear counterpart. When 4 nt loops are ranked according to their UI values, this sequence is 18th for both A3B-full and A3B-CTD, but is 7th for A3A (Supplementary Data 1). Hence it seemed possible that this hairpin was not preferred strongly over the linear sequence because it is not an intrinsically preferred by A3B. We tested this possibility using a biochemical assay for A3A and A3B-CTD (Fig. 4A). To determine whether cytosines in hairpin loops were better substrates for A3B than linear substrates we compared the activity of purified A3B-CTD on a linear substrate with a GTT.C. sequence and several hairpins with TT.C. in their loops.

A hairpin loop with GTT.C. sequence was in fact a much worse substrate than four other hairpins including the one containing TTT.C. (Fig. 4B). However, in contrast with the previous report[40] we found that the hairpin with GTT.C. sequence was a better substrate for A3B-CTD than its linear counterpart (Fig. 4C). The reasons for the difference with the Buisson et al. paper[40] are unclear, but we note

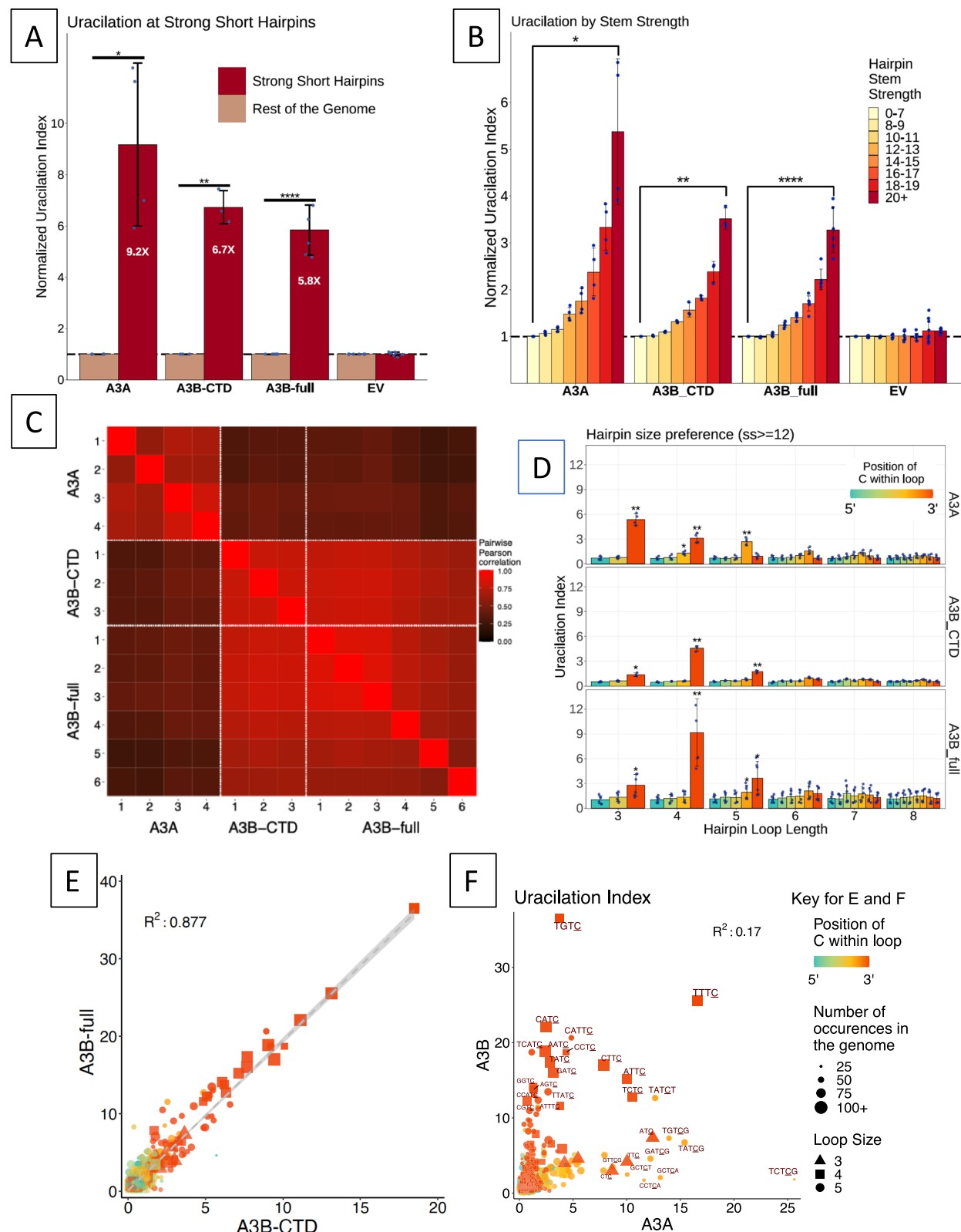

that in a more recent publication from the Buisson group[52], the authors found that the same hairpin was indeed a better substrate for A3B than its linear counterpart (Fig. 1E in that paper). Additionally, hairpins with TT.C. and AGTT.C. sequences were much better substrates than the GTT.C. linear (Fig. 4C). These results confirm the finding of UPD-seq experiments that A3B-CTD prefers hairpin loop substrate over an extended chain.

It is difficult to compare purified A3A and A3B-CTD directly with each other because depending on the purity and stability of each batch, the amount of active enzyme in each preparation can be different. To overcome this difficulty, we normalized the activity of the enzymes against the activity for the loop sequence (TTT.C.) which is a good substrate for both enzymes according to the UI values (Fig. 2F). When the activity of A3A and A3B-CTD was compared for hairpins with

**Fig. 2 | Preferences of A3A and A3B for cytosines in hairpin loops. A** Uracilation of hairpins with short loops (less than 6 nt) with stable stems (SS ≥ 15) for A3A, A3B-CTD, and A3B-full are compared with the empty vector (EV) control. The uracilation index (UI) was normalized to the value of genome-wide UI. Bar heights and whiskers respectively represent mean and standard deviation. A paired two-sided t-test compared the A3A, A3B-CTD, or A3B-full against the EV values and the P-values were corrected for multiple testing hypotheses. The exact P-values are: A3A −0.01423; A3B-CTD- 0.00423; A3B-full- 6.6×10⁻⁵. **B** Dependance of normalized UI on stem strength of hairpins. Bar heights and whiskers respectively represent mean and standard deviation. A paired two-sided t-test compared UI values for the highest and lowest interval of stem strength hairpin loops. The exact P-values are: A3A − 0.01109; A3B-CTD- 0.002869; A3B-full- 9.2 × 10⁻⁵. **C** Pairwise comparison of different UPD-seq samples. The comparison is based on uracilation within hairpin loops. **D** Preferences of A3A, A3B-CTD and A3B-full for cytosines in different loop sizes and at different positions within the loops. Bar heights and whiskers respectively represent mean and standard deviation. The statistical significance was

determined in the following way- A one-way ANOVA test was performed to determine if significant difference exists between the samples in each loop length-loop position group. A post-hoc t-test (unpaired, two-sided t-test) was performed on significant groups from the ANOVA test, comparing each of the A3A, A3B-CTD, or A3B-full against EV samples. These P-values were then adjusted for multiple comparisons using Benjamini-Hochberg method. The exact P-values are reported in the Source Data File for this figure. In parts (**A**) through (**D**), the bars with high statistical significance are marked with "*" (P ≤ 0.05), "**" (P ≤ 0.01), "***" (P ≤ 0.001) or "****" (P ≤ 0.0001). **E.** Correlation between UI of A3B-CTD and A3B-full for hairpin loops of different sizes and loop sequences. Only loops with 3, 4, or 5 nt are shown. **F.** The UI values of A3B-full and A3A hairpin loops for different loop sequences. Only loops with 3, 4, or 5 nt are shown. The color scheme for the position of cytosine within the loop, symbols for loops of 3, 4, and 5 nt, and size for the occurrence of each loop sequence in the genome for both (**E**) and (**F**) are presented to the right of part (**F**). Bar heights and whiskers respectively represent mean and standard deviation. Source data are provided as a Source Data file.

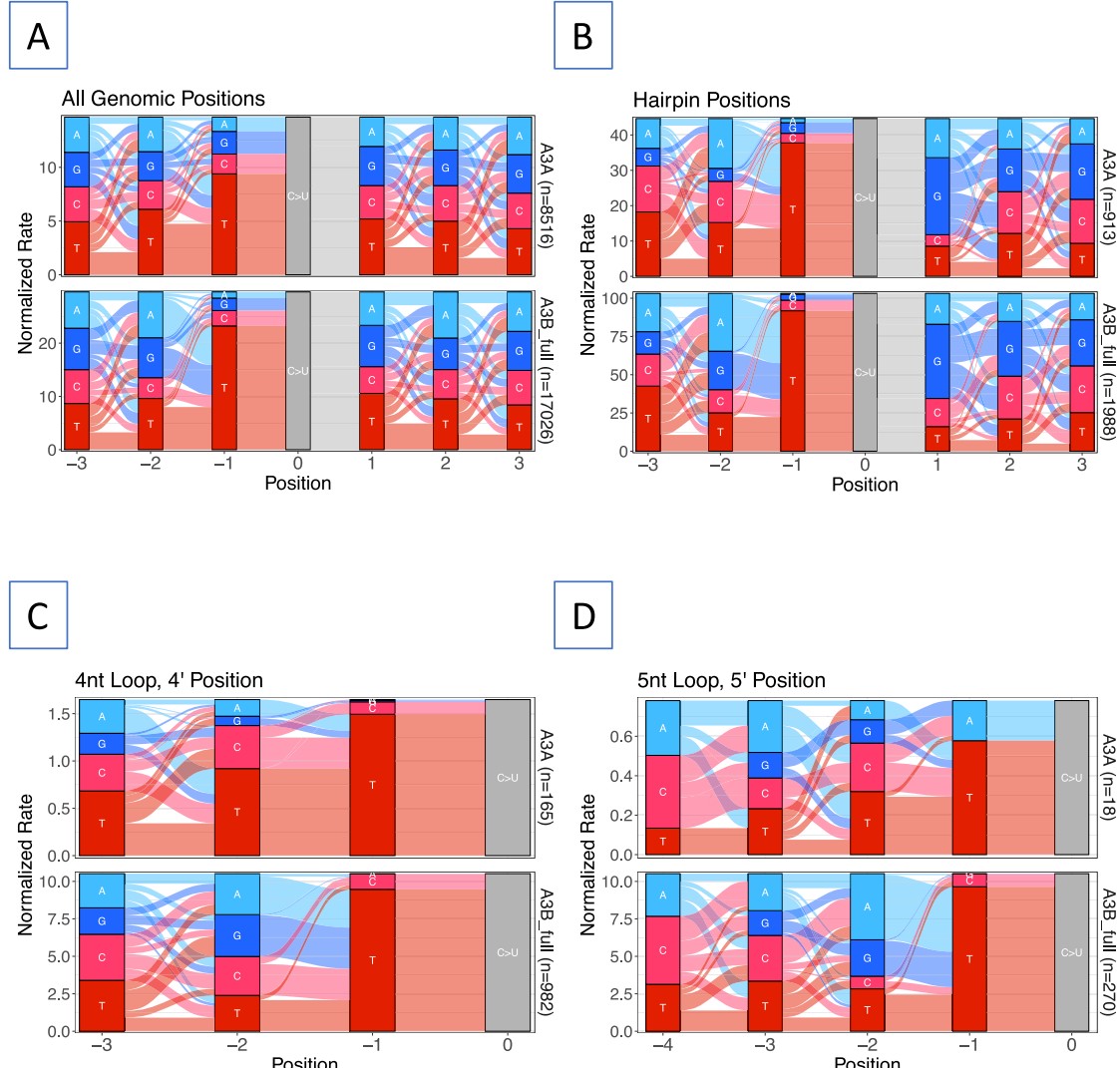

**Fig. 3 | Sequence context of deaminated cytosines.** The normalized rate of different base sequences upstream and downstream of the deaminated cytosine (marked "0") at all genomic position with UI > 0.04 was displayed as River plots[48]. The width of the river from between two bases is proportional to the frequency of

occurrence of the two bases next to each other normalized to the available sequences in the genome. **A** All genomic cytosines; **B** Cytosines in all positions of 3 nt through 6 nt hairpins (SS ≥ 10); **C** 4 nt loops with cytosine at 3′ end (4/4); **D** 5 nt loops with cytosine(C) at 3′ end (5/5). Source data are provided as a Source Data file.

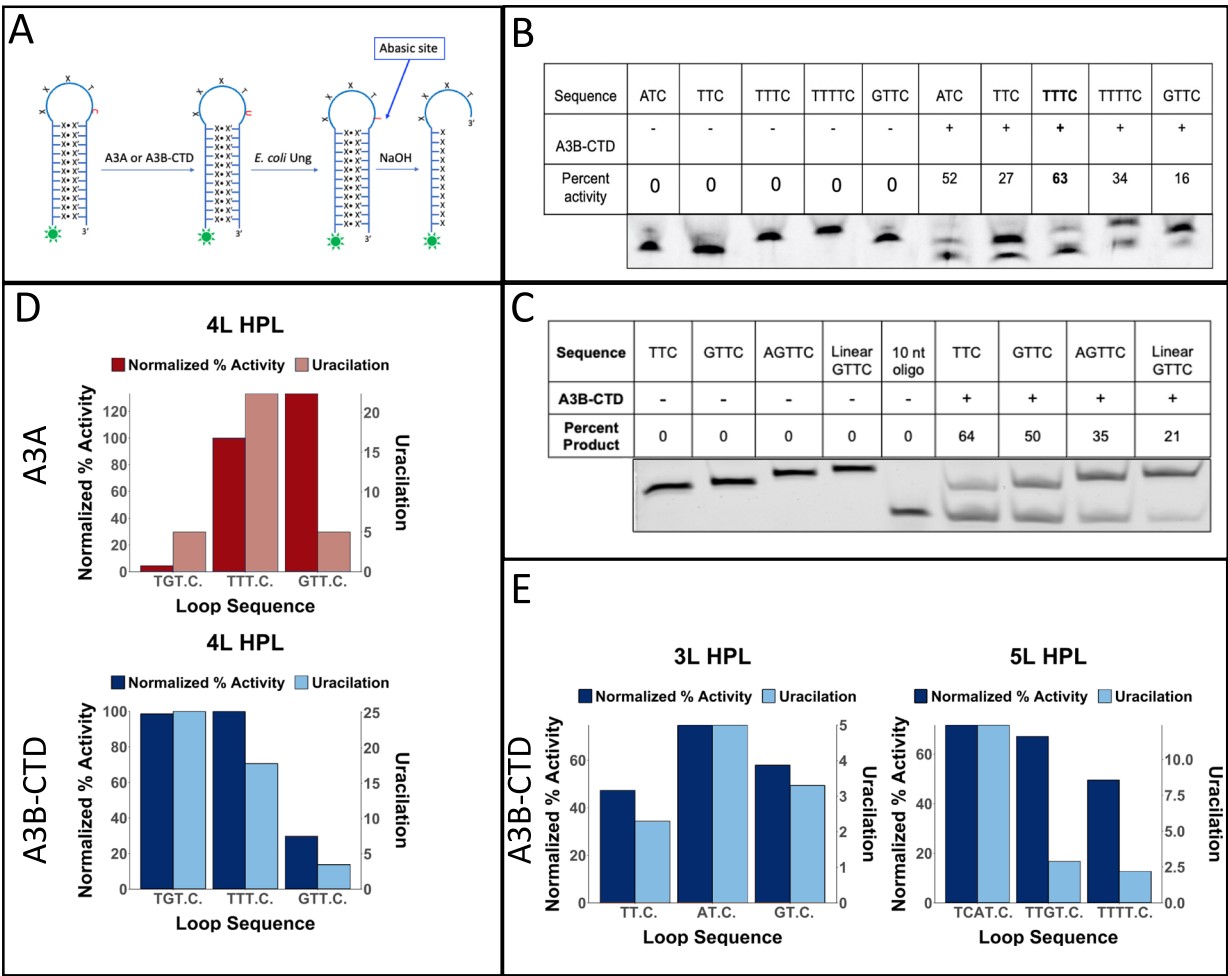

**Fig. 4 | Biochemical activity of A3A and A3B-CTD for different hairpin loops.**
**A** Schematic representation of cytosine deamination assay. A hairpin DNA with a specific loop sequence is synthesized with a 6-FAM label at 5′ end. The DNA is incubated with either A3A or A3B-CTD followed by E. coli Ung. The resulting abasic site is cleaved using NaOH to create a shorter fluorescent DNA molecule. The substrate DNA is separated from the product using electrophoresis through a denaturing gel. **B** Comparison of biochemical activity of A3B-CTD for a hairpin with a GTTC loop and hairpins containing other loop sequences. The oligomers were incubated with the enzyme for 10 min. **C** Comparison of the activity of A3B-CTD for a linear oligonucleotide and several hairpins containing TTC in the loop. The incubation time was 20 min. The activity assays for each DNA oligomer were performed three or more times. One representative gel picture is shown. All the data sets and the original gel images used for parts (**B**) and (**C**) are included in the Source Data File. In parts (**B**) and (**C**), the symbol "-" in rows labeled A3B-CTD indicate absence of the enzyme in the reaction, while symbol "+" represents presence of the enzyme in the reaction. **D**. Comparison of biochemical activities and UI values of A3A and A3B-CTD for hairpins for three different 4 nt loops. The biochemical activity data are normalized for activity for hairpin with TTTC.
**E** Comparison of biochemical activities and UI values of A3B-CTD for hairpins for different 3 nt and 5 nt loops. The biochemical activity data are normalized for activity for hairpin with TTTC. In parts (**D**) and (**E**), Uracilation is uracilation index, UI. Source data are provided as a Source Data file.

three different tetranucleotide sequences using TTT.C. for normalization, GTT.C. was better substrate for A3A than for A3B-CTD (Fig. 4D; and Supplementary Fig. S8). Additionally, we tested a hairpin loop with TGT.C. sequence because hairpin loops in the *E. coli* genome with this sequence have much higher UI for A3B, than for A3A (Fig. 2F). This is consistent with the normalized biochemical activities of A3A (Fig. 4D, upper panel) and A3B-CTD (Fig. 4D, lower panel).

We also determined the activity of A3A and A3B-CTD for different 3 nt and 5 nt loop sequences containing T.C. dinucleotide and compared them to the UI values of the sequences in UPD-seq (Fig. 4E and Supplementary Fig. S8). For trinucleotide loops, A3B-CTD showed the highest enzyme activity and UI value for the sequence AT.C., while showing the lowest values for both the parameters, for TT.C. Similarly, TCAT.C. was the best pentanucleotide substrate for A3B-CTD and had the highest UI value among the tested substrates. A3B-CTD showed lowest activity for TTTT.C. and this sequence also had the lowest UI value as well (Fig. 4E). When the twelve biochemical activity results and the corresponding UI values shown in Fig. 4D, E were subjected to

Spearman correlation test, the Spearman coefficient was 0.74 and the P-value was 0.006 demonstrating that there is a general correlation between enzyme activity against hairpin loop substrates and their UI values in UPD-seq.

## Tumor genome mutations in loops have stronger A3A characteristics than A3B

We examined the human tumor whole genome sequence datasets from a combination of The Cancer Genome Atlas (TCGA) and the International Cancer Genome Consortium's Pan-Cancer Analysis of Whole Genomes (PCAWG) project[16] to determine whether the preferences of A3A and A3B for different hairpin loop sizes and positions were detectable in tumor mutations. Specifically, we selected tumors with the most A3A-like character (A3A-most; mutations in YTCA context » RTCA context) or with the most A3B-like character (A3B-most; mutations in RTCA context » YTCA context) according to the criteria defined by ref. 39, and asked whether their mutations showed loop preferences of A3A and A3B see in UPD-seq.

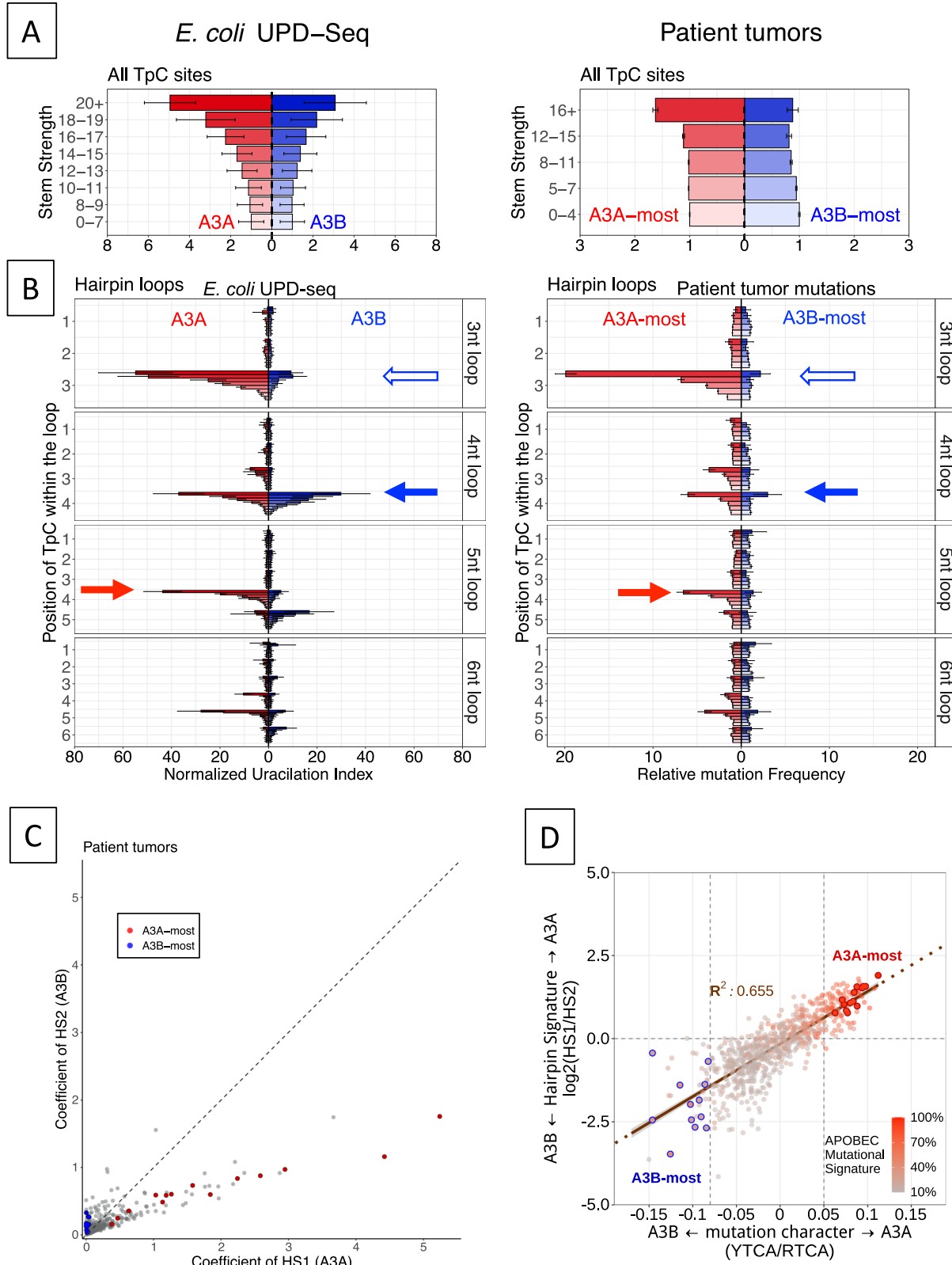

In Fig. 5A, left panel, we have replotted the uracilation in *E. coli* caused by A3A and A3B in a more compact form than shown in Figs. 2B, D. Both A3A and A3B show a strong dependence on stem strength (Fig. 5A, left panel). When these data are stratified by loop length and position of cytosine within the loop, A3A and A3B respectively show a preference for 3 nt and 4 nt loops and A3A shows a preference for cytosines in position 4 within 5 nt loops (red arrow,

Fig. 5B, left panel). When APOBEC signature mutations in TC sequences in A3A-most or A3B-most human tumors were analyzed in a similar way, A3A-most tumors readily show the characteristic dependance of stem strength, a preference for 3 nt loops (Fig. 5A, B right panels), and a preference for position 4 in 5 nt loops (closed red arrow; Fig. 5D, right panel). These data show that the A3A-most tumors reflect the intrinsic preferences of the enzyme for stem strength, loop size, and cytosine

**Fig. 5 | Properties of human tumor mutations in hairpin loops. A** Side-by-side comparison of uracilation index due to A3A and A3B in *E. coli* (left panel) and normalized mutation rate in A3A-most and A3B-most patient tumors shown with increasing hairpin-forming stem strength (right panel). Both UI and mutation rates are calculated at TpC sequences. The error bars show confidence intervals based on the total number of mutations observed. **B** Side-by-side comparison of UI in *E. coli* and normalized mutation rate in A3A-most and A3B-most patient tumors at hairpins with different loop lengths and loop positions with increasing stem strength. The number of A3A-most and A3B-most tumors were respectively 12 and 14. The error bars show confidence intervals based on the total number of mutations observed. **C** Coefficient values assigned to each sample for HS1 and HS2 are plotted

on the x and y axis. Each point is a tumor sample. Samples that were selected as A3A-most and A3B-most are highlighted using red and blue colors. The dotted line is y = x. **D** Log2 ratio of coefficients of HS1 and HS2, obtained from NMF-based hairpin signature analysis, against the mutation character from the −2 position preference. The correlation between the two metric is shown via the brown line. Only samples with fraction of APOBEC mutations more than 10% are used in this plot. The blue to red colors show percentage of APOBEC mutational signature. Vertical dashed lines at −0.08 and 0.05, show the threshold originally used to select A3A- and A3B-most samples used in parts (**A**, **B**). A3A- and A3B-most samples are highlighted as larger red and blue dots. Source data are provided as a Source Data file.

position as seen in both *E. coli* uracilome and in biochemical properties of the enzyme.

In contrast, the intrinsic hairpin loop preferences of A3B were harder to discern in the A3B-most tumors. When hairpins of different loop sizes were put together, the mutation frequency did not show a dependence on stem strength (Fig. 5A, right panel). However, when the hairpins were separated by loop size, cytosine position and stem strength, a pattern emerged at highest stem strength. A3B-most tumors had more mutations at position 3 in 3 nt loops (3/3) and position 4 in 4 nt loops (4/4) (open and closed arrows, respectively, Fig. 5B, right panel) with the highest number of 4/4 (closed arrow Fig. 5B, right panel) and they were highest at highest stem strength. This is the same pattern seen with the uracilation created by A3B in *E. coli* (Fig. 5B, left panel). Based on this similarity, we conclude that some tumors with a high percentage of mutations in RTCA sequences do reflect intrinsic properties of the A3B protein, but in a much weaker way than how the loop preferences of A3A are reflected in tumors with a high percentage of mutations in YTCA sequences.

### Analysis of human tumor mutations using A3A and A3B hairpin signatures

Mutational signatures based on trinucleotide motifs reveal distinct mutagenic processes underlying a sample's mutations. COSMIC mutational signatures 2 and 13, attributed to APOBEC3 activity[10], are unable to distinguish between the activity of A3A and A3B. We created two signatures HS1 and HS2 that respectively reflect the loop size and cytosine position preferences of A3A and A3B (Supplementary Fig. S2). We applied these signatures to the human tumor mutations from the TCGA and PCAWG datasets assigning an HS1 and HS2 coefficients to each tumor. The plot is shown in Fig. 5C.

When the A3A vs. A3B character of each tumor as represented by Log2(ratio HS1/HS2) is plotted against the motif-based mutation character (YTCA/RTCA), the points can be fitted to a straight line with R2-value 0.655 (Fig. 5D). The plot shows that tumors with a high percentage of APOBEC signature mutations predominantly contain A3A signature mutations (Fig. 5D). Furthermore, the A3A-most and A3B-most tumors separately cluster together suggesting that these two independent ways of defining relative contribution of A3A and A3B to tumor mutations (i.e., HS1/HS2 and RTCA/YTCA ratios) are roughly equivalent. Thus the hairpin characteristics of mutations generated by A3A and A3B provide information that is orthogonal to that obtained from nucleotide sequence motifs[39].

## Discussion

### A3B-CTD is responsible for the substrate selectivity by A3B

A3B-CTD is the catalytic part of A3B enzyme, but it was unclear prior to this study whether A3B-NTD played any role in substrate selection. The results of this study show that the NTD plays little role in this process. A3B-CTD and A3B-full behaved similarly to each other in terms of the genomic regions they targeted (Fig. 1A), their preference for LGST over LDST (Fig. 1B), hairpin loops over linear DNA (Fig. 2A) and 4 nt loops over loops of other sizes (Fig. 2D). In fact, there was a strong correlation between loop sequences preferred by A3B-CTD and A3B-full

(Fig. 2E) and the preferences of the two enzymes at the −1 and −2 positions (Supplementary Data 1). The only consistent difference between A3B-CTD and A3B-full was that A3B-full expression resulted in a higher number of uracilation peaks (Fig. 1A), and when comparing the same loop sequences it displayed higher UI values (Supplementary Data 1). It seemed possible that this difference between the two proteins could be due to higher expression of the latter protein in *E. coli* or intrinsically higher activity for A3B-full. To distinguish between these possibilities we performed Western blot analysis of *E. coli* cell extracts expressing the two proteins. Unexpectedly, we found that full-length protein is expressed at four- to eight-fold higher levels than it catalytic domain (Supplementary Fig. S9). Together, these results are consistent with a model in which active A3B enzyme interacts with its preferred substrates exclusively through its CTD, and the NTD increases its activity by structural means such as increasing enzyme stability or speeding its search for a suitable substrate.

### A3A and A3B have similar but distinguishable, preferences for hairpin loops

The results of both UPD-seq of *E. coli* expressing A3B-full or A3B-CTD (Fig. 2A) and biochemical assays using purified A3B-CTD (Fig. 4A) show that A3B has a strong preference for cytosines in short, stable hairpin loops. While the overall preference for such loops across the *E.coli* genome was about five-fold for A3B-full (Fig. 2A), A3B-CTD had a three-fold preference for one hairpin with a trinucleotide loop compared to the corresponding linear sequence (Figs. 4B, C). This result was unexpected because a previous study[40] had reported that another hairpin (loop sequence GTT.C.) was only as good a substrate for A3B in cell-free extracts as its linear form. It is likely that this happened because loops with GTT.C. are much poorer substrates for A3B compared to A3A (Figs. 4B, C, D).

Although both the enzymes prefer hairpin loop sequences over their linear counterparts, there are significant differences in the preferences of A3A and A3B in terms of loop size and sequence. Specifically, A3A prefers trinucleotide loops the best, whereas both A3B-CTD and A3B-full prefer tetranucleotide loops over other loop sizes (Fig. 2D). This was also seen in biochemical assays using A3B-CTD, where a TT.C. containing hairpin was the best substrate for A3A (Supplementary Fig. S8), but a TTT.C.-containing hairpin was the best substrate for A3B (Fig. 4B and Supplementary Fig. S8). We have previously noted that the A3A protein has the shortest loop 1 sequence among the AID/APOBEC family and shown that replacing it with the longer sequence from AID leads to a change in preference for 3 nt hairpin loops to 4 nt loops[44]. The loop 1 of A3B-CTD and AID have the same length, but different sequences and hence these results suggest that the size of the nucleotide loops preferred by the deaminase has a direct relationship with the size of loop 1 in the protein.

There were other differences in the selection of sequences by A3A and A3B, the most notable being the preferred position for target cytosine in 5 nt loops. Whereas the 4th position was preferred by A3A (4/5), the 3' end position within the loop (5/5) was preferred by both A3B-CTD and A3B (Fig. 2D). This was reflected in the rankings of pentanucleotide loop sequences based on UI values (Table 1). The top

## Table 1 | Ranking of pentanucleotide sequences

| | A3A | | | A3B-full | | |
|---|---|---|---|---|---|---|
| | Loop position | Loop Sequence | UI | Loop position | Loop Sequence | UI |
| 1 | **4** | TCT.C.G | 25.59 | 5 | CATT.C. | 20.64 |
| 2 | **4** | TAT.C.G | 15.37 | 5 | TCAT.C. | 18.72 |
| 3 | **4** | TGT.C.G | 13.92 | 5 | TTAT.C. | 13.47 |
| 4 | **4** | GCT.C.A | 13.11 | 5 | CCAT.C. | 13.02 |
| 5 | **4** | TAT.C.T | 12.63 | **4** | TAT.C.T | 12.73 |
| 6 | **4** | GAT.C.G | 12.19 | 5 | ATTT.C. | 12.35 |
| 7 | **4** | CCT.C.A | 11.59 | **4** | CAT.C.C | 11.53 |
| 8 | **4** | GCT.C.T | 10.02 | 5 | CAAT.C. | 11.19 |
| 9 | **4** | GTT.C.G | 8.67 | 5 | AAAT.C. | 10.97 |
| 10 | **4** | TTT.C.G | 7.91 | 5 | TTGT.C. | 10.49 |
| 11 | **4** | GAT.C.A | 7.83 | 5 | CGGT.C. | 9.72 |
| 12 | **4** | AAT.C.A | 5.71 | 5 | CTAT.C. | 9.70 |
| 13 | **4** | CGT.C.A | 5.51 | 5 | ACAT.C. | 9.62 |
| 14 | **4** | AAT.C.G | 5.32 | 5 | CCGT.C. | 9.15 |
| 15 | **3** | TT.C.GT | 5.09 | 5 | CGAT.C. | 9.07 |
| 16 | **4** | TTT.C.T | 4.87 | **4** | GGT.C.T | 8.47 |
| 17 | 5 | CATT.C. | 4.83 | **3** | GT.C.CG | 8.30 |
| 18 | **4** | GTT.C.A | 4.81 | 5 | ATAT.C. | 8.23 |
| 19 | **4** | GAT.C.T | 4.64 | 5 | TAAT.C. | 8.19 |
| 20 | **4** | ATT.C.G | 4.59 | **4** | TGT.C.G | 7.38 |

Sequences with cytosine in any position other than 5, are in **bold**.
*UI* uracilation index.

20 pentanucleotide sequences by UI value for A3A contain 18 that have the cytosine at position 4, whereas this is true for only 5 out of the 20 top sequences for A3B (Table 1).

Neither enzyme showed a strong preference for a specific base at the +1 position when all cytosines were considered (Fig. 3A) and this is consistent with the conclusions of a yeast mutational study[39]. However, when the yeast mutations were examined in the TC context, YTCA was the preferred sequence for A3A, while RTCA was preferred by A3B[39]. There was also a modest preference for G in the +1 position when mutations were restricted to the TC context[39], and this is not seen in our data (Fig. 3A, not shown). It is possible that each model organism (yeast and *E. coli*) has a different composition of hairpin loop and other sequences that affects these results. There may also be unrecognized biases in the two experimental set-ups that may skew the results. Interestingly, in our data both enzymes prefer a G at +1, +2, and +3 positions when hairpins are considered and the frequency of (G + C) » (A + T) at these positions (Fig. 3B). It is likely that this happens because these positions represent bases in the last three base-pairs in the stem. G:C pairs near the closing end of the stem would make the stem more stable.

### Human tumor mutations predominantly show loop preferences of A3A, not A3B

Whereas the cancer genomes mutations identified to have a strong A3A-like character by the criteria outlined by ref. 39 do clearly display the intrinsic secondary structure preferences of the A3A enzyme, the same is not true of mutations in tumors with supposedly strong A3B-like character (Fig. 5A, B, C). Unlike the A3A-most tumors, the A3B-most tumor mutations as a whole show no dependence on stem strength of the hairpins (Fig. 5A). The intrinsic properties of A3B show up when the mutations in A3B-most tumors are separated based on a combination of loop size, cytosine position, and stem strength, but only in a dampened form.

We created two hairpin signatures, HS1 and HS2, based on the C to U conversion in loops of 3, 4, 5, and 6 nt by respectively A3A and A3B

(Supplementary Fig. S2) and applied the metric to assess the APOBEC mutational history of tumors in the TCGA and PCAWG datasets. This method ignores nucleotide sequence and is blind to the −2 position underlying the well-known RTCA/YTCA preference[39]. The method correlates significantly with the motif-based method and correctly classifies the A3A-most and A3B-most tumors (Fig. 5D).

### Hypothesis about why A3B hairpin mutational signature is diminished in human tumors

It is possible that A3A displays its preferences in *E. coli*, yeast, mice (see accompanying manuscript by ref. 52), and human mutations without alterations because its action is not assisted or hindered by other cellular proteins, whereas the mutagenic action of A3B is hindered by protein(s) in native human cells. These inhibitory proteins may be absent in heterologous hosts including *E. coli* and yeast, and their inhibitory actions may not be as strong in a murine host cell that does not code for endogenous A3A or A3B proteins.

The idea that activity of A3B, but not A3A, is diminished in human cells makes biological sense. While A3A is expressed "episodically" in cancer cell lines, A3B is expressed constitutively in many cancer lines[19,35,53]. Additionally, A3B is overwhelmingly located in nuclei, while A3A appears to shuttle between the nuclei and the cytoplasm[26–29]. Thus, A3B has much greater potential of causing mutations in dividing cells than A3A. One way in which the cells thwart A3B from causing mutational mischief is through the binding of RNA to its NTD thereby making the enzyme much less active and this is one of the arguments used to explain why A3B is unlikely to be the source of cancer genome mutations with APOBEC signature[18].

We suggest here that RPA may also play a role in blocking A3B from causing mutations. As mentioned above, both A3A and A3B preferentially cause mutations in the LGST presumably because it is more often single-stranded than the LDST. We have speculated previously that this ssDNA can form hairpin loops that are the true targets for A3A and A3B. As RPA (in mammalian cells) or SSB (in *E. coli*) also bind ssDNA in LGST, there is a competition between RPA/SSB and A3A/A3B for binding the hairpin loops[43,45]. If RPA/SSB binds the ssDNA, it would prevent A3A/A3B from deaminating cytosines. It appears that SSB in *E. coli* is unable to prevent A3A or A3B from binding to hairpins in LGST, but the human RPA successfully prevents A3B, but is less efficient in blocking A3A from causing deaminations. It is possible that A3B contains sequences that allow RPA to interact with the protein in such a way that it can displace it from ssDNA. As the principal difference between A3A and A3B is the existence of NTD in the latter protein, it is tempting to suggest that RPA interacts with A3B through the NTD of the latter protein to prevent this enzyme from causing hazardous mutations.

## Methods

### Bacterial strains and plasmids

*E. coli* strains including BH214 (GM31 *ung⁻ mug⁻*) and BL21(DE3) were respectively used for the expression of either A3B-CTD or A3B-full, and the purification of A3B-CTD and A3A proteins. Human A3B-CTD and A3B-full were separately cloned into pASK-IBA5C plasmid vector (IBA LifeSciences) using Gibson cloning kit (Thermo Fisher Scientific). The clones were confirmed using colony PCR followed by the digestion of plasmid DNA using appropriate restriction enzymes and by Sanger sequencing (Genomics Core, Michigan State University). A3B-CTD was also cloned into pET28a plasmid vector for protein purification purposes and its sequence was also confirmed using Sanger sequencing.

### Optimization of A3B-CTD expression in *E. coli* for UPD-seq

To optimize induction of A3B-CTD protein in *E. coli*, BH214, cells containing pASK-IBA5C-A3B-CTD were grown in LB medium containing chloramphenicol (25 µg/mL) at 37 °C overnight. Following overnight growth, the cultures were diluted 50-fold using LB media, and subjected to shaking at 37 °C for 2.5 h. The cultures were diluted 10X

and split into two for induced and un-induced cultures. For A3B expression in induced samples, different concentrations of AHT (from Cayman Chemical) 0.01 μg/mL, 0.025 μg/mL, 0.05 μg/mL and 0.1 μg/mL were added. Cells were further grown to O.D. of 0.8 and collected using centrifugation at 4 °C. Cells were broken using sonication in cell lysis buffer (20 mM Tris pH8 and 50 mM NaCl). Cell-free lysate was obtained by centrifuging cells to get rid of cellular debris in pellet. 10 μg of cell extract was loaded into SDS-PAGE gel (15% w/v) and transferred to PVDF membrane (Millipore Corp.). After transferring proteins to PVDF, membranes were blocked using 5% wt./v skimmed milk. Membrane was subjected to rabbit anti-human A3B monoclonal antibody (1:500 dilution, NIH AIDS Reagent Program, catalog number 5210-87-13) followed by goat anti-rabbit IgG H&L (HRP). Proteins were visualized using super signal west pico plus (Thermo Fisher Scientific) chemiluminescence substrate and signal was detected using FluorChemQ (Cell Biosciences Inc.) gel scanner.

### Expression and purification of A3B-CTD and A3A proteins from *E. coli*

BL21(DE3) cells containing pET28a-A3B-CTD were grown to mid-log phase at 37 °C and IPTG was added to 0.4 mM to induce transcription of the A3B gene. The cells were further grown overnight at 18 °C and spun down by centrifugation (Beckman Coulter). A french press (Thermo Spectronic) was used to break the cells in the cell lysis buffer [20 mM Tris-HCl (pH 8), 50 mM NaCl] supplemented with cOmplete™ protease inhibitor cocktail (Sigma Aldrich) and cell-free lysate was collected after centrifugation to remove cell debris. The lysate was applied to the Ni-NTA agarose resin (Qiagen) and the resin was washed three times using a buffer [20 mM Tris-HCl (pH 8), 50 mM imidazole] containing successively higher concentrations of NaCl (50 mM, 250 mM, and 500 mM, respectively). The bound protein was eluted using elution buffer [20 mM Tris-HCl (pH 8), 50 mM NaCl, 250 mM imidazole]. The protein was loaded onto Superdex-75 increase column (AKTA GE FPLC) and the eluted fractions were analyzed on a SDS-PAGE gel (Supplementary Fig. S1A). The fractions containing purified protein were pooled together and were concentrated to 10 μg/μL in storage buffer [20 mM Tris-HCl (pH 8), 50 mM NaCl, 10% glycerol, 1 mM DTT, and 1 mM EDTA] and stored at −80 °C. A3A was purified from BL21DE3 cells containing pET-28a-A3A plasmid using essentially the same procedure except protein expression was induced at 0.5 mM IPTG (Supplementary Fig. S1B).

### Western blot analysis of A3B-CTD and A3B-Full

Expression of A3B-CTD or A3B-full was induced in BH214 cells for 5 h using anhydrotetracycline (AHT) at concentrations of 0.05 μg/mL or 0.1 μg/mL. The proteins were separated on SDS-PAGE and the proteins were detected using rabbit anti-APOBEC3B monoclonal antibody (1:500 dilution, NIH AIDS Reagent Program, catalog number 5210-87-13). A3B-CTD purified from *E. coli* served as a positive control.

### UPD-seq of cells expressing A3B-CTD or A3B-full

The UPD-seq procedure described previously in ref. 44,46 was followed with minor variations. The genomic DNA was sonicated to create fragments of 500 bp size using Covaris M220 instrument. The fragmented DNA was treated with 10mM O-allylhydroxylamine (AA7) for 1 h at 37 °C to block pre-existing abasic sites. This DNA was treated with the *E. coli* uracil-DNA glycosylase (Ung, 5 units) for 30 min at 37 °C to excise genomic uracils and further incubated with 2 mM ssARP for 1 h at 37 °C. The DNA was purified to remove ssARP and applied to activated MyOne Dynabeads Streptavidin C1 magnetic beads (Invitrogen). The DNA bound to the magnetic beads was separated from unbound DNA on a magnetic stand (DynaMag, Invitrogen) and the beads were washed five times using 1X bind and wash buffer. The bound DNA was released from the beads using 100 mM DTT for 10 min at 37 °C. The DNA was concentrated using ethanol precipitation and the pellet was dissolved in 0.1XTE buffer. The concentration of DNA was determined

using the Nano-drop 2000c instrument (Thermo Scientific) or the Qubit 4 fluorometer (Thermo Fisher Scientific) with 1X dsDNA High Sensitivity and Broad Range assay kit (Invitrogen). This pulled-down DNA was used to prepare libraries for sequencing using TruSeq DNA nano library preparation kit according to standard protocols with the exception that the Taq polymerase was used during the library amplification step. The resulting DNA libraries were checked for quality and purity using Qubit and Bioanalyzer (Agilent 2100), and subsequently pooled, denatured, and sequenced using Illumina Mini-Seq instrument (Department of Biological Sciences, Wayne State University).

### Biochemical activity assays for A3A and A3B-CTD

Activity assays for purified A3B-CTD or A3A were performed with 6-FAM labeled gel purified oligonucleotide as substrates in a reaction buffer [20 mM Tris-Cl (pH 8), 1 mM DTT, 1 mM EDTA] at 37 °C for the indicated length of time. The oligonucleotide sequences are provided in Supplementary Table S3. The reactions were terminated by heating samples for 10 mins at 95 °C. Oligos were treated with *E. coli* Ung for 30 mins at 37 °C. Reactions were terminated by adding NaOH to 0.1 M and heating the solutions for 10 mins at 95 °C. The reaction products were separated on denaturing polyacrylamide gel (15% w/v) and the DNA products were visualized and photographed using iBright FL 1500 (Thermo Fisher Scientific). The band intensities in the gels were determined using ImageJ software (NIH).

### Bioinformatics analysis

All raw sequencing data was transferred from BaseSpace to HPC Grid (Wayne State University) using Globus. LINUX software available on the HPC grid was used to perform Bioinformatics analysis using raw FASTA sequence data files. BWA (version 0.7.12) was used to map and then exclude reads that aligned to plasmid DNA. Remaining reads were aligned to the *E. coli* reference genome using the same tool. Aligned reads output was used to extract depth of coverage using Samtools (version 1.2.83).

### Normalized differential coverage version 2 (NDC2)

The NDC2 method was implemented to determine uracil peaks against genome as described previously in ref. 46. A threshold of $5\sigma$ was used to call true uracil peaks and normalization was done using the average depth of coverage. R studio (version 3.4.1) ggplot2 was used to create plots and figures. The genes and other genomic elements that overlapped uracilation peaks were determined using the BLAST alignment function within EcoCyc database[54] and mapped to the *E. coli* MG1655 genomic sequence. Hairpin prediction within *E. coli* was performed as described previously in refs. 44,46.

### Calculation of uracilation Index

Uracilation Index (UI) is the frequency at which any cytosine in the genome is converted to uracil. It is the mean fraction of C to T or G to A changes at a specified nucleotide position or a genomic region normalized to the depth of coverage. This number is multiplied by 1000 for convenience of handling.

$$UI = \sum \left( \frac{Number\ of\ C\ to\ T\ or\ G\ to\ A\ changes}{Depth\ of\ coverage} \right) \times 10^3 \qquad (1)$$

### Pairwise comparison of UPD-seq datasets

We conducted a comprehensive analysis of the uracilation at hairpins with loop lengths between 3 nt to 6 nt, considering nucleotide sequence of the hairpin loops, and the position of the cytosine within the loop. A table was created with all possible combinations. Applying a minimum stem strength of 10 to select for hairpin-forming sequences. Stem strength is defined by the number of base-pairs at the stem sequence (ss = 3X G:C pairs + 1X A:T pairs). We calculated the

uraciation index (UI) for each combination and sample. We visualized the relationships between samples using a heatmap plot, which shows the pairwise correlations computed from the UI values.

## ANOVA analysis of UI data for cytosines in different sized loops

A one-way ANOVA analysis of UI data was performed under the assumption of unequal variance to compare the sample groups and find loop length and positions that are different among the groups. After adjusting the *P*-values from this step to correct for multiple testing hypotheses, we performed a post-hoc t-test comparing the UI values at each of these hairpin sizes. The t-test compared the A3A, A3B-CTD, or A3B-full against the EV values. Once again, the *P*-values were corrected for multiple testing hypotheses. The bars with high statistical significance are marked with "*" ($P \leq 0.05$) or "**" ($P \leq 0.01$).

## Construction of the river plots

For each set of targeted cytosines, we gathered all the cytosines with a minimum uraciation fraction of 0.04 (to remove noise) and a maximum uraciation fraction of 0.8 (to remove clonal mutations) and created GRange objects[55]. We adapted and modified several functions from the MutationalPatterns R package[51] to normalize the number of uraciated positions by the total number of available genomic sites with the same sequence. The river plots were generated to visualize the normalized transition probabilities and the distribution of nucleotides surrounding a uraciated cytosine.

## Analysis of human tumor mutation data

We selected from the TCGA + PCAWG dataset[16], tumor samples based on the presence of APOBEC3 mutational signature as determined by non-negative matrix factorization[11], and identified them as A3A-like or A3B-like based on the fraction of APOBEC mutational signature and the ratios of YTCA/RTCA mutations[39,41]. A3A-most and A3B-most tumors were described previously as most-A3A and most-A3B[40].

Non-negative matrix factorization (NMF) is a dimensionality reduction method commonly used to detect mutational signatures in cancer genomes[11]. Using the uracil deamination patterns observed at hairpins in *E. coli* samples expressing human A3A or A3B, we designed a NMF-based method, operating on hairpin characteristics instead of trinucleotide motifs, to characterize the activity of the two enzymes and evaluate their activity levels in human cancer genomes. First, we counted the number of deaminated TpC positions (with uraciation fraction $\geq 0.03$) in each set of A3A- and A3B-expressing *E. coli* samples and grouped them based on hairpin characteristics: the size of the potential hairpin loop (3–6 nt), the position of the deaminated cytosine within the loop and the stem strength (binned into 5 groups: 1–4, 5–7, 8–11, 12–15 and 16+). We thus obtained two vectors which we define as Hairpin Signature 1 (HS1) from A3A samples, and Hairpin Signature 2 (HS2) from A3B samples (Supplementary Fig. S2). We similarly counted mutations in human tumors and grouped them by hairpin characteristics, creating a matrix of mutation counts. We then used the two hairpin signatures, HS1 and HS2, to factorize the mutation count matrix by directed NMF. This gives each sample a pair of coefficients for each of the two hairpin signatures. A small constant, 0.03, was added to the coefficients prior to calculating the Log2 ratio. We have created MATALB functions to streamline the hairpin signature analysis and have made them available on GitHub along with the rest of the code.

## Reporting summary

Further information on research design is available in the Nature Portfolio Reporting Summary linked to this article.

## Data availability

The raw sequence files generated in this study are deposited at NCBI-SRA BioProject PRJNA1005650. Source data are provided with this paper.

## Code availability

The source code for the software used is available at https://github.com/rayanramin/APOBEC3B-UPDSeq.

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

## Acknowledgements

We are grateful to Dr. Jared Schrader (Department of Biological Sciences, Wayne State University) for the use of Illumina MiniSeq Sequencer. We would also like to thank Jessica Stewart (Wayne State University) for the purification of human A3A and to Rémi Buisson (University of California, Irvine) for communicating results prior to publication. This work was supported by National Institutes of Health [1R21AI144708 and 1R21CA252858-01A1] and a Bridge Funding grant from Wayne State University (to A.S.B.) and a Rullo Family Innovation Award (to M.S.L.).

## Author contributions

Y.B., R.S. and R.M.-R. performed laboratory experiments, analyzed the data and wrote parts of the manuscript. R.S. and M.L. did computational analysis of the experimental data and analysis of tumor mutations. A.B. conceived the project, guided the research, and wrote the manuscript.

## Competing interests

The authors declare no competing interests.
