## [Peer Review File · Nature Communications]

Distinguishing preferences of human APOBEC3A and APOBEC3B for cytosines in hairpin loops, and reflection of these preferences in APOBEC-signature cancer genome mutationsEditorial Note: Parts of this Peer Review File have been redacted as indicated to remove third-party material where no permission to publish could be obtained.

REVIEWER COMMENTS

Reviewer #1 (Remarks to the Author):

In this manuscript, Butt, et al investigate the propensity for APOBEC3B to act at DNA stem loops (hairpins) using uracil pull-down sequencing (UPD-seq). The related enzyme, APOBEC3A, is well described to act at stem loops but previous data has suggested that APOBEC3B does not act at stem loops. The authors use an e.coli system in which human APOBEC enzymes are expressed to define the locations at which cytosines are converted to uracils, indicative of enzyme activity. Since both APOBEC3A and B are suspected to act as mutagens in cancer, the authors sought to distinguish mutational patterns of each enzyme. Their major finding is that APOBEC3B mutates cytosines in stem loops with four bases, whereas APOBEC3A mutates cytosines in stem loops with three bases. While minor differences in mutation substrate are uncovered by this research, the impact of the findings are relatively minor. The authors associate their findings in e. coli to mutational patterns in human cancers. Stem loop activity of each enzyme defined in e. coli is analyzed in human cancers, although the consistency of these findings is unclear. Much of the data are supportive of prior publications (i.e., Fig 1B replication bias, Fig 2 data regarding A3A activity on stem loops, Fig 3 sequence preferences of A3A and A3B). I appreciate the value of reproducibility especially in a different experimental system (e. coli) using a unique way to identify APOBEC mutations (UPD-seq), however the novelty of findings in this manuscript are incremental. In addition, several points of clarification and additional analyses are required to strengthen the data:

1. The authors compare A3B-CTD with full length A3B and conclude that the N-term domain of A3B does not impact substrate preference (lines 266-267 and 360-361, 370-371). However, this conclusion is inconsistent with the data given that full length A3B generates >2x more uracil peaks than A3B-CTD. While the CTD might direct substrate preference moreso than the NTD, the data do not support the model presented.

2. Recurrent mutations at the same genomic position are found not only across biological replicates but also between A3B-CTD and full length A3B. This is curious given the stochastic nature of APOBEC enzymes and the lack of recurrent hotspot mutations that have been described previously in the literature.

3. There is substantial discordance between uracilation index and biochemical activity on a synthetic substrate in Fig 4. Specifically 4C A3A graph GTT.C sequence, 4D 5L HPL graph TTGT.C and TTTT.C sequences. This calls into question the validity of using uracil index to define substrate preference. In addition for Fig 4, the terms 3L, 4L etc should be defined in the figure or legend and it would be helpful to show the full hairpin substrate for all panels.

4. I had a very hard time understanding Fig. 5A. Pentaloops were relatively disfavored by A3B compared to A3A (based on data in Supp Table 2) so using a 5-base loop as the distinguishing feature between A3A and A3B activity seems biased against detecting A3B activity. In the figure, it appears that a small fraction of APOBEC high tumors have a 4/5 ratio >3 whereas zero APOBEC low tumors have a ratio >3. How can this be interpreted? The authors state that the ratio is overwhelmingly greater than 1 (line 422) but that interpretation should be backed by statistical analysis rather than description. Statistics are also needed for Fig 5B where it appears that A3B-most and APOBEC-low have a similar ratio of hairpin mutations. Fig 5C and 5D also seem to show that there are very few TpC mutations and/or very few mutated stem loops in the A3B-most tumors, which is inconsistent with data in e. coli. In addition for Fig 5: the colors in 5A are hard to see. Figure legend is missing for 5D. It would be helpful in the legend to define where the data come from and to state how the APOBEC mutation signature was defined. Also 3/3 and 4/4 hairpins should be defined in the legend.

Additional minor suggestions:

Line 53-54 the authors suggest that base changes alone define the APOBEC mutational signature but this is misleading. SBS signatures are defined by base change and trinucleotide context.

The discussion of AID in the introduction (lines 107-112) seems irrelevant in the context of the manuscript.

Supplementary Figure 2 requires more explanation in the legend to understand the figure.

Line 287 says Supplementary twice

Line 416-417 it would be helpful to define in the text which human tumor WGS database was interrogated

Fig 3: while I understand the rationale for using River plots and the data that can be determined from this type of analysis, the images in Fig 3 are extremely difficult to interpret. Perhaps the authors could include a visual summary of the river plot analysis for ease of interpretation by readers.

Reviewer #2 (Remarks to the Author):

Butt and Colleagues exploit a recently developed assay for uracil detection in DNA to analyze the differences in targeting of APOBEC3A and APOBEC3B (A3A and A3B), two deaminases involved in the onset of cancer mutations. The Authors perform these experiments in E.coli expressing A3B, either wild type or its catalytic domain, and compare the uracil patterns induced by A3B in the bacterial genomes with those induced by A3A (from a previous work). Through this comparison, the Authors determine the favored sequence contexts and stem-loop preference for both A3B and A3A - the most notable being the position of the mutated C within the hairpin. These preferences are then confirmed by biochemical assays. Finally, the Authors show that most tumors with the APOBEC mutational signature present the targeting characteristics of A3A, while tumors with A3B-derived mutations present less apparent signatures.

Despite the relative simplicity of the experimental design, the manuscript represents a step forward in our understanding of the contribution of these two enzymes to the mutational milieu in human cancer. Indeed, its strength lies in the unbiased approach used to characterize the characteristics that target APOBEC3A and APOBEC3B to the DNA. Any other approach would have not have allowed to identify the discriminating preference for targeting cytosines at different positions within a loop.

Major comments

- Overall, the manuscript is difficult to read as there is a general lack of structure (e.g. the analysis in the human tumors is only mentioned in the Discussion, when it would also belong to the Results; the logic behind the selection of hairpins to be tested is not provided -Fig. 4C- or is explained after presenting the results -Fig. 4D). Considering how straightforward the work is, it might be better to combine Results and Discussion together.
- Despite the assay to determine the uracilome has been already published, it would help the reader a proper introduction at the beginning of the Results.
- Considering that the analysis relies mostly on a single, unbiased approach (genome-wide analysis of uracilome), it is important to make sure hidden biases are not present. In particular, it is not clear whether the normalization described in the Methods to calculate the Uracilation Index regards the full hairpin sequence (loop in the context of stems of a given strength) or only the loop portion (no matter the context). Based on the Supplementary Files, I would say that, at least for position -1, a bias is not apparent but, for farther positions, a careful normalization might be needed.
- Analogously, since a single approach is used, it would be better if the results are supported by some statistical analysis.

Minor comments

- Considering the similarity in the approach (genome-wide analysis of APOBEC3A/B targeting in a non-mammalian system), it might be useful to discuss the findings also in light of Taylor et al. (2013, 10.7554/eLife.00534)
- Figure 4A. While adequate for the specialist, the deamination assays would benefit from a better representation (e.g. a schematic representation of the assay; the activity shown close to the labels can be confounding: the "-" sign is a label in the upper row and a value in the lower one)

Reviewer #3 (Remarks to the Author):

In this manuscript the authors investigated APOBEC3B (A3B) substrate preference, both in terms of 2°

structure and in terms of sequence specificity. Authors extended the current knowledge base for APOBEC3B biochemical substrate specificity by showing that it displays a strong preference for hairpin-forming (vs linear) ssDNA in an UNG-deficient strain of *E. coli* (uracilation assays). This preference was also observed in human tumors by examining sequence context of APOBEC mutations and by differentiating A3B- from APOBEC3A (A3A)-derived mutations using previously reported preferences for deaminating cytidines preceded by RT and YT respectively (where Y is a pyrimidine and R is a purine). Authors were able to further differentiate A3B substrate specificity from that of A3A in *E. coli* using in vitro deaminase activity assays where A3B exhibited a preference for larger loops of 4 and 5nt, with the target cytidine in the most 3' position (A3A instead preferring C in the penultimate position of 5nt loops, and generally preferring smaller loop sizes, i.e. 3 nt as has been previously reported) and showed that, taking loop size and cytidine positioning into account, this preference is mediated by hairpin stability as calculated from stem length and composition. It is noteworthy that the authors reporting of A3B's preference for hairpins contradicts a previous report that indicated it preferred linear. The authors cite differences in hairpin loop composition and the potential for unrecognized biases in previous studies as potential reasons for this contradiction.

The authors also examined uracilation in *E. coli* expressing full length A3B or just the C-terminal domain (compared to previous uracilation data from A3A expression), observing that the N-terminal domain of A3B has little influence on A3B deaminase activity or substrate specificity, with both full-length and truncated A3B showing a comparable preference tRNA genes and ssDNA generated on the lagging strand during replication. The authors noted increased and broader activity for full-length A3B in *E. coli*, suggesting that substrate specificity is largely conferred by A3B's C-terminal domain, and they posited that the N-terminal domain increases overall activity through increased enzyme stability or by increasing the rate at which A3B can access substrate (no data provided but consistent with prior reports).

This manuscript advances the field of APOBEC cancer biology by further characterizing A3B substrate specificity and provides a method for better distinguishing mutations caused by off-target activity from A3B versus A3A. The methodologies used and experiments were adequately rigorous, with reasoned conclusions being made. However, beyond the finding that A3B in fact does prefer hairpins the results are an incremental advance and may be more suitable for a specialized journal focused on nucleic acid modifying enzymes. The manuscript is generally well written. My specific comments follow:

1. A3B-CTD activity and full length A3B have very similar sequence specificities, but the authors indicate that full length A3B is more active in *E. coli* (lines 365-367). A western showing comparable levels of expression of the two enzymes needs to be provided to support this claim as differences in protein expression could account for this.
2. The authors identify deamination sequence preference for both A3A and A3B as a G at the +1 position and indicate this is different from previous results in yeast (Lines: 403-412). However, the results in yeast that are cited indicate preference for both enzymes of G or A at the +1 position (<https://www.nature.com/articles/ng.3378>; Figure 1), which is pretty consistent with the current data. The usage of TCA as "preference" in cited manuscript is for defining the differential Y or R nucleotide in the -2 position for A3A or A3B, respectively. The authors of the cited paper most strongly see the -2 nucleotide preference in the context of a TCA containing tetranucleotide. This section should be clarified.
3. The authors suggest the reason that A3B was previously indicated to prefer linear over hairpin sequences was do to the choice of hairpin loop sequence (lines 380-383). They indicate this is supported by poorer activity of A3B on GTTC loops compared to other loop sequences. The authors should compare the GTTC loop to a GTTC linear sequence to make this claim. The data in figure 4A actually indicates that GTTC loops are better substrates than TTC linear substrates for A3B which would indicate this is not the correct reason for the difference with the previous report (<https://www.science.org/doi/10.1126/science.aaw2872>)
4. While the authors see evidence of A3B activity at hairpins in human tumors, this activity is less obvious than for A3A and no case is made for a larger utility of this information beyond understanding

the enzymology of A3B. Is there any evidence of A3B hairpin specificity being important for driver mutations, recurrent mutations, or could this information be used to better distinguish tumors experiencing A3B mutations versus A3A mutations?

Response to Reviewer Comments

We wish to thank all the reviewers for their thoughtful comments and suggestions for the improvement of the manuscript. We have made great efforts to improve the manuscript based on these comments and the revised manuscript is attached. In particular, we made additional calculations to show statistical significance of different comparisons and added supplementary figures to better explain our computational and experimental methods. In addition, we changed slightly the Tables 1 and S3 to make them more consistent with the computational analysis underlying Figure 2. Whereas this figure was generated using stem strength (ss) ≥ 10 for hairpins, the two tables were generated using ss > 10 . The new Tables use ss ≥ 10 , but this does not change the ranking of different loop sequences. We have also corrected an error we made in the calculation of standard deviation for the uracilation index (UI, Table S3). This also does not affect any of the conclusions reported in the text.

Reviewer #1

Reviewer #1 (Remarks to the Author):

In this manuscript, Butt, et al investigate the propensity for APOBEC3B to act at DNA stem loops (hairpins) using uracil pull-down sequencing (UPD-seq). The related enzyme, APOBEC3A, is well described to act at stem loops but previous data has suggested that APOBEC3B does not act at stem loops. The authors use an e.coli system in which human APOBEC enzymes are expressed to define the locations at which cytosines are converted to uracils, indicative of enzyme activity. Since both APOBEC3A and B are suspected to act as mutagens in cancer, the authors sought to distinguish mutational patterns of each enzyme. Their major finding is that APOBEC3B mutates cytosines in stem loops with four bases, whereas APOBEC3A mutates cytosines in stem loops with three bases. While minor differences in mutation substrate are uncovered by this research, the impact of the findings are relatively minor. The authors associate their findings in e. coli to mutational patterns in human cancers. Stem loop activity of each enzyme defined in e. coli is analyzed in human cancers, although the consistency of these findings is unclear. Much of the data are supportive of prior publications (i.e., Fig 1B replication bias, Fig 2 data regarding A3A activity on stem loops, Fig 3 sequence preferences of A3A and A3B). I appreciate the value of reproducibility especially in a different experimental system (e. coli) using a unique way to identify APOBEC mutations (UPD-seq), however the novelty of findings in this manuscript are incremental. In addition, several points of clarification and additional analyses are required to strengthen the data:

1. The authors compare A3B-CTD with full length A3B and conclude that the N-term domain of A3B does not impact substrate preference (lines 266-267 and 360-361, 370-371). However, this conclusion is inconsistent with the data given that full length A3B generates $>2x$ more uracil peaks than A3B-CTD. While the CTD might direct substrate preference more so than the NTD, the data do not support the model presented.
2. Recurrent mutations at the same genomic position are found not only across biological replicates but also between A3B-CTD and full length A3B. This is curious given the stochastic nature of APOBEC enzymes and the lack of recurrent hotspot mutations that have been described previously in the literature.
3. There is substantial discordance between uracilation index and biochemical activity on a synthetic substrate in Fig 4. Specifically 4C A3A graph GTT.C sequence, 4D 5L HPL graph TTGT.C and TTTT.C sequences. This calls into question the validity of using uracil index to define substrate preference. In addition for Fig 4, the terms 3L, 4L etc should be defined in the figure or legend and it would be helpful

to show the full hairpin substrate for all panels.

4. I had a very hard time understanding Fig. 5A. Pentaloops were relatively disfavored by A3B compared to A3A (based on data in Supp Table 2) so using a 5-base loop as the distinguishing feature between A3A and A3B activity seems biased against detecting A3B activity. In the figure, it appears that a small fraction of APOBEC high tumors have a 4/5 ratio >3 whereas zero APOBEC low tumors have a ratio >3 . How can this be interpreted? The authors state that the ratio is overwhelmingly greater than 1 (line 422) but that interpretation should be backed by statistical analysis rather than description. Statistics are also needed for Fig 5B where it appears that A3B-most and APOBEC-low have a similar ratio of hairpin mutations. Fig 5C and 5D also seem to show that there are very few TpC mutations and/or very few mutated stem loops in the A3B-most tumors, which is inconsistent with data in *e. coli*. In addition for Fig 5: the colors in 5A are hard to see. Figure legend is missing for 5D. It would be helpful in the legend to define where the data come from and to state how the APOBEC mutation signature was defined. Also 3/3 and 4/4 hairpins should be defined in the legend.

Additional minor suggestions:

Line 53-54 the authors suggest that base changes alone define the APOBEC mutational signature but this is misleading. SBS signatures are defined by base change and trinucleotide context.

The discussion of AID in the introduction (lines 107-112) seems irrelevant in the context of the manuscript.

Supplementary Figure 2 requires more explanation in the legend to understand the figure.

Line 287 says Supplementary twice

Line 416-417 it would be helpful to define in the text which human tumor WGS database was interrogated

Fig 3: while I understand the rationale for using River plots and the data that can be determined from this type of analysis, the images in Fig 3 are extremely difficult to interpret. Perhaps the authors could include a visual summary of the river plot analysis for ease of interpretation by readers.

Author response

Response to general comments:

A. "Their major finding is that APOBEC3B mutates cytosines in stem loops with four bases, whereas APOBEC3A mutates cytosines in stem loops with three bases."

Response: While the preference for 3 nt loops (A3A) vs. 4 nt loops (A3B) is one important result from this study, it is not the most important one. The most important result is that contrary to a previous report (1,2) which reported that A3A -- but not A3B -- prefers cytosines in hairpin loops, our work shows that both the enzymes have a preference for hairpin loops. This raises the very important question of why hairpin TC mutations in cancer genomes with a high percentage of APOBEC signature mutations overwhelmingly have A3A character (YTC context) and lack A3B character (RTC context) (Fig. 3B in the paper (2)). We confirm and extend this observation to show that even in tumors with a very high percentage of A3B character mutations ("A3B-most"), hairpin loop mutations show only a very weak preference for 4 nt loops (Fig. 5D). Thus, even among A3B-most tumors, most mutations attributable to A3B occur in non-loop sequences. We address this paradox in the Discussion.

B. " While minor differences in mutation substrate are uncovered by this research, the impact of the findings are relatively minor."

Response: We note that there are important biological differences between the various APOBEC enzymes. The two enzymes restrict human viruses with different efficiencies and may play different roles in carcinogenesis (3). Being able to clearly distinguish the mutation signatures of the different APOBEC enzymes is crucial for a full understanding of their roles in various diseases, including cancer, where A3A and A3B are the main contributors to APOBEC

mutagenesis. Our findings shed light on the question of the relative contributions of these two enzymes in cancer (Fig. 5). Our data clearly show that A3A makes much larger contribution to these mutations than A3B. We now speculate in the Discussion about the likely reason why A3B makes a smaller contribution to hairpin loop mutations than A3A and have added a sentence in the Abstract to emphasize this point.

C. "Stem loop activity of each enzyme defined in *e. coli* is analyzed in human cancers, although the consistency of these findings is unclear."

Response: We do not understand what the reviewer means by "consistency of these findings", especially because in the next sentence s/he says that "Much of the data are supportive of prior publications".

D. "...however the novelty of findings in this manuscript are incremental."

Response: As pointed out in response to reviewer's comment B, this manuscript addresses key questions regarding the roles of A3A and A3B in cancer genome mutations.

Response to specific comments:

1. The authors compare A3B-CTD with full length A3B and conclude that the N-term domain of A3B does not impact substrate preference (lines 266-267 and 360-361, 370-371). However, this conclusion is inconsistent with the data given that full length A3B generates >2x more uracil peaks than A3B-CTD. While the CTD might direct substrate preference more so than the NTD, the data do not support the model presented.

Response: We thank the reviewer for this comment, which highlights the difference between the large-scale and small-scale aspects of APOBEC substrate targeting. We wish to point out that the generation of a uracilation peak and the selection of specific 3 nt, 4 nt, 5 nt loop substrates (i.e. substrate preference or specificity) are distinct concepts. The uracilation peaks presumably occur in regions of the genome that have persistent single-stranded DNA. These are typically a couple of hundred base pairs long, and occur predominantly around transcription start sites (TSS; Supplementary Tables S1 and S2 and Supplementary Fig. S3). However, A3B-full generates more peaks than A3B-CTD because of higher expression levels in *E. coli* (See below our response to comment #1 by Reviewer #3). The substrate specificity of the two enzymes is nearly identical as shown by Fig. X2E. To make this distinction clearer, we have changed the sentence in lines 266-267 to (Lines 287-289) "These results also suggest that the N-terminal domain of A3B does not have a strong influence on the **types** of genomic regions preferentially targeted by A3B" (Change in **red**).

2. Recurrent mutations at the same genomic position are found not only across biological replicates but also between A3B-CTD and full length A3B. This is curious given the stochastic nature of APOBEC enzymes and the lack of recurrent hotspot mutations that have been described previously in the literature.

Response: The reviewer is confusing sequencing reads of uracil-containing fragments with mutations. DNA fragments with uracils were covalently tagged, pulled-down and sequenced. When these sequencing reads were mapped to the *E. coli* genome, the reads piled up in some regions such as tRNA genes and TSS. In most cases, the positions of uracils in different overlapping sequencing reads were not identical. Also, unlike in a mutational study, where typically only a few mutations may be found per clone, we are simultaneously mapping thousands of uracils per genome in millions of cells. We have now added a graphical explanation of uracilation peaks, as Supplementary Fig. S2. Furthermore, APOBEC mutation hotspots do occur. Buisson *et al.* showed that passenger mutations in cancer genomes with

APOBEC signature occur frequently in hairpin loops (1,4) and we see a similar accumulation of uracils in hairpin loops in the *E. coli* genome following expression of either A3A or A3B.

3. There is substantial discordance between uracilation index and biochemical activity on a synthetic substrate in Fig 4. Specifically 4C A3A graph GTT.C sequence, 4D 5L HPL graph TTGT.C and TTTT.C sequences. This calls into question the validity of using uracil index to define substrate preference. In addition for Fig 4, the terms 3L, 4L etc should be defined in the figure or legend and it would be helpful to show the full hairpin substrate for all panels.

Response: We are not suggesting that the uracilation index (UI) of cytosines in different sequence contexts has a perfectly linear relationship with *in vitro* biochemical activity of A3A and A3B-CTD. This is highly unlikely because in cells access of A3A/A3B to cytosines in single-stranded DNA is constrained by the double-stranded nature of DNA and physiological factors such as transcription and binding of proteins to DNA. However, we show that the trends in UI and biochemical activity are similar. To establish this point, we performed spearman correlation tests between the 12 pairs of biochemical activity and UI values. The text in lines 370-371 of the original manuscript now reads- (Lines 378-382) "When the twelve biochemical activity results and the corresponding UI values shown in Fig. X4C and X4D were subjected to Spearman correlation test, the Spearman coefficient was 0.594 and the P-value was 0.046 demonstrating that there is a general correlation between enzyme activity against hairpin loop substrates and their UI values in UPD-seq."

4. I had a very hard time understanding Fig. 5A. Pentaloops were relatively disfavored by A3B compared to A3A (based on data in Supp Table 2) so using a 5-base loop as the distinguishing feature between A3A and A3B activity seems biased against detecting A3B activity. In the figure, it appears that a small fraction of APOBEC high tumors have a 4/5 ratio >3 whereas zero APOBEC low tumors have a ratio >3. How can this be interpreted? The authors state that the ratio is overwhelmingly greater than 1 (line 422) but that interpretation should be backed by statistical analysis rather than description. Statistics are also needed for Fig 5B where it appears that A3B-most and APOBEC-low have a similar ratio of hairpin mutations. Fig 5C and 5D also seem to show that there are very few TpC mutations and/or very few mutated stem loops in the A3B-most tumors, which is inconsistent with data in *e. coli*. In addition for Fig 5: the colors in 5A are hard to see. Figure legend is missing for 5D. It would be helpful in the legend to define where the data come from and to state how the APOBEC mutation signature was defined. Also 3/3 and 4/4 hairpins should be defined in the legend.

Response: APOBEC3B does not target pentaloops less than APOBEC3A. The mean UI value among different pentaloop positions for A3B-full is 1.68 (median 0.98) and for A3A is 1.09 (0.65). More significantly, within pentanucleotide loops A3A tends to deaminate cytosines at position 4 more frequently (mean UI = 2.46) than those at position 5 (0.87), while A3B-full has the opposite preference (mean UI at 4 and 5 positions are 1.74 and 3.03, respectively). We have now changed the relevant text (Lines 440-444) to read- (Lines 385-388) "As noted above, a major difference between A3A and A3B pertains to preferred position of cytosine in hairpins with pentaloops. A3A tends to deaminate cytosines at position 4 more frequently (mean UI = 2.46) than those at position 5 (0.87), while A3B-full has the opposite preference (mean UI at 4th and 5th positions are 1.74 and 3.03, respectively)."

Additionally, we have now normalized the mutations at the 4th and 5th position of the pentaloops to the number of available sequences in the genome. This removes any statistical bias in the analysis of the data. Finally, if mutations were not caused by a discriminatory mechanism such as A3A or A3B (low % of APOBEC mutations), we expect no enrichment, i.e. a normalized ratio near 1. However, among the tumor samples we have analyzed, there is an enrichment of mutations at the 4th position (4/5) relative to the 5th position (5/5) in hairpins. This increased ratio or enrichment is high when the mechanism that caused it is more active. We

have now added new text at the end of that paragraph- (Lines- 397-400) "This shows that tumor samples with higher percentage of APOBEC signature mutations are more likely to deviate from the null ratio of 1 and this deviation is suggestive of a history of mutagenesis mostly driven by A3A. Using a one-sample t-test with the null hypothesis 4/5 mutation ratio of 1, we calculate a P-value $< 2.2 \times 10^{-16}$ indicative of very high significance."

We apologize for the missing legend for Fig. 5D. The legend has now been added. We have also clarified the terms 3/3 and 4/4 as "three nucleotide hairpins with cytosine at 3rd position (3/3)" and "four nucleotide hairpins with cytosine at 4th position (4/4)", respectively.

Response to minor suggestions:

Line 53-54 the authors suggest that base changes alone define the APOBEC mutational signature but this is misleading. SBS signatures are defined by base change and trinucleotide context.

The discussion of AID in the introduction (lines 107-112) seems irrelevant in the context of the manuscript.

Response: The relevant sentence has been changed to- (Lines 54-57) "Error-prone repair or replication of these uracils creates predominantly C:G to T:A or G:C mutations, in TC dinucleotides within specific trinucleotide contexts (generally TCN, where N is any nucleotide) and are referred to as APOBEC signature mutations"

Supplementary Figure 2 requires more explanation in the legend to understand the figure.

Response: The figure legend has been changed to- "Salmon colored boxes are uracilation peaks created by A3B-CTD and A3B-full and *E. coli* genes with three letter name abbreviations are light green boxes. Experimentally determined transcription start sites (TSS) are vertical lines with left- or right-pointing arrows. The gene positions and TSS are according to EcoCyc database*."

Line 287 says Supplementary twice

Response: This has been corrected.

Line 416-417 it would be helpful to define in the text which human tumor WGS database was interrogated

Response: The relevant sentence now says (Lines 388-391) "We examined the human tumor whole genome sequence datasets from a combination of The Cancer Genome Atlas (TCGA) and the International Cancer Genome Consortium's Pan-Cancer Analysis of Whole Genomes (PCAWG) project..."

Fig 3: while I understand the rationale for using River plots and the data that can be determined from this type of analysis, the images in Fig 3 are extremely difficult to interpret. Perhaps the authors could include a visual summary of the river plot analysis for ease of interpretation by readers.

Response: We have now added a sentence in the text- (Lines 326-328) "We chose river plots instead of the more common sequence logo plots because the former are able to capture relational information between enriched bases (Supplementary Fig. S6)." and we now explain the advantage of river plots over SeqLogo plots using a fictional example in the supplementary figure S6.

Reviewer #2 (Remarks to the Author):

Butt and Colleagues exploit a recently developed assay for uracil detection in DNA to analyze the differences in

targeting of APOBEC3A and APOBEC3B (A3A and A3B), two deaminases involved in the onset of cancer mutations. The Authors perform these experiments in E.coli expressing A3B, either wild type or its catalytic domain, and compare the uracil patterns induced by A3B in the bacterial genomes with those induced by A3A (from a previous work). Through this comparison, the Authors determine the favored sequence contexts and stem-loop preference for both A3B and A3A - the most notable being the position of the mutated C within the hairpin. These preferences are then confirmed by biochemical assays. Finally, the Authors show that most tumors with the APOBEC mutational signature present the targeting characteristics of A3A, while tumors with A3B-derived mutations present less apparent signatures.

Despite the relative simplicity of the experimental design, the manuscript represents a step forward in our understanding of the contribution of these two enzymes to the mutational milieu in human cancer. Indeed, its strength lies in the unbiased approach used to characterize the characteristics that target APOBEC3A and APOBEC3B to the DNA. Any other approach would have not have allowed to identify the discriminating preference for targeting cytosines at different positions within a loop.

Major comments

- Overall, the manuscript is difficult to read as there is a general lack of structure (e.g. the analysis in the human tumors is only mentioned in the Discussion, when it would also belong to the Results; the logic behind the selection of hairpins to be tested is not provided -Fig. 4C- or is explained after presenting the results -Fig. 4D). Considering how straightforward the work is, it might be better to combine Results and Discussion together.
- Despite the assay to determine the uracilome has been already published, it would help the reader a proper introduction at the beginning of the Results.
- Considering that the analysis relies mostly on a single, unbiased approach (genome-wide analysis of uracilome), it is important to make sure hidden biases are not present. In particular, it is not clear whether the normalization described in the Methods to calculate the Uracilation Index regards the full hairpin sequence (loop in the context of stems of a given strength) or only the loop portion (no matter the context). Based on the Supplementary Files, I would say that, at least for position -1, a bias is not apparent but, for farther positions, a careful normalization might be needed.
- Analogously, since a single approach is used, it would be better if the results are supported by some statistical analysis.

Minor comments

- Considering the similarity in the approach (genome-wide analysis of APOBEC3A/B targeting in a non-mammalian system), it might be useful to discuss the findings also in light of Taylor et al. (2013, 10.7554/eLife.00534)
- Figure 4A. While adequate for the specialist, the deamination assays would benefit from a better representation (e.g. a schematic representation of the assay; the activity shown close to the labels can be confounding: the “-” sign is a label in the upper row and a value in the lower one)

Author response

Response to general comments:

We thank the reviewer for pointing out the simplicity and unbiased nature of our approach and the importance of our results to cancer genome mutations.

Response to specific comments:

Major comments

- Overall, the manuscript is difficult to read as there is a general lack of structure (e.g. the analysis in the human tumors is only mentioned in the Discussion, when it would also belong to the Results; Considering how straightforward the work is, it might be better to combine Results and Discussion together.

Response: We apologize for any confusion caused by the order of presentation of tumor mutation data. We have now moved those data into a new section of the Results. We discuss the implications of these results in the Discussion.

- the logic behind the selection of hairpins to be tested is not provided -Fig. 4C- or is explained after presenting the results -Fig. 4D).

Response: We have described in the sentences immediately before the discussion of activity assays why we chose to test a hairpin with the tetranucleotide loop -(Lines 351-353) "In a previous report(1) it was noted that several hairpin sequences were poorer substrates for A3B

than A3A. In particular, a hairpin with GTT.C. as the loop sequence was only as good a substrate as its linear counterpart." Later, we have explained why we normalized the data with respect to the activities of the enzymes on a substrate with a TTT.C. loop. A hairpin with this sequence has a high UI value for both the enzymes (Fig. 2F). We have now added sentences to explain why we chose TGT.C. as the third loop sequence for comparison- (Lines 368-371) "Additionally, we tested a hairpin loop with TGT.C. sequence because hairpin loops in the *E. coli* genome with this sequence have much higher UI for A3B, than for A3A (Fig. 2F). This is consistent with the normalized biochemical activities of A3A (Fig. 4C, left panel) and A3B-CTD (Fig. 4C, right panel)."

- Despite the assay to determine the uracilome has been already published, it would help the reader a proper introduction at the beginning of the Results.

Response: The technology is explained in the penultimate paragraph of the Introduction. We explain the methodology further in that section: (Lines 103-109) "We developed an experimental methodology which has addressed these questions using the *E. coli* genome as the target for these enzymes. Following expression of one of the AID/APOBEC family of enzymes in an UNG⁻ strain of *E. coli*, the genomic DNA is extracted and deoxyuridines are converted to abasic sites. The abasic sites are reacted with a chemical containing a detachable biotin. The tagged DNA fragments are pulled down using streptavidin beads, released from the beads, and are sequenced and mapped to the bacterial genome (5). This strategy, which is termed uracil pull-down sequencing (UPD-seq)..." Additionally, the first sentence in the Results has been modified as follows: (Lines 264-266) "The genomic distribution of uracils, the uracilome, created by full-length A3B (A3B-full) and the carboxy-terminal domain of A3B (A3B-CTD) was determined in multiple independent samples using UPD-seq (see Introduction above; (5)) and..."

- Considering that the analysis relies mostly on a single, unbiased approach (genome-wide analysis of uracilome), it is important to make sure hidden biases are not present. In particular, it is not clear whether the normalization described in the Methods to calculate the Uracilation Index regards the full hairpin sequence (loop in the context of stems of a given strength) or only the loop portion (no matter the context). Based on the Supplementary Files, I would say that, at least for position -1, a bias is not apparent but, for farther positions, a careful normalization might be needed.

Response: We have taken the issue of statistical significance very seriously throughout our analysis. When measuring the UI at any group of hairpin loop sequences, we normalize the values to the total number of genomic sites matching that hairpin sequence which are within the range of stem strength we are analyzing. For example, in Figures 2E and 2F, the UI value of (TGTC) which is a preferred substrate by A3B, but not A3A, is normalized by measuring the uracilation amount in A3A and A3B samples normalized to the number of (TGTC) hairpins in the genome.

- Analogously, since a single approach is used, it would be better if the results are supported by some statistical analysis.

Response: We thank the reviewer for this suggestion, and accordingly we have now performed ANOVA analysis of the data Figs. 2B and 2D and noted the statistical significance above the bars in these figure. We have also noted the R² values for the plots 2E and 2F within the figures. The Spearman coefficients show that whereas UI values for A3B-CTD and A3B-full have a strong correlation, the same is not true for the correlation between A3A and A3B-full. The analysis procedure is explained in Materials and Methods as follows: (Lines- 232- 239)

"ANOVA analysis of UI data for cytosines in different sized loops

A one-way ANOVA analysis of UI data was performed under the assumption of unequal variance to compare the sample groups and find loop length and positions that are different

among the groups. After adjusting the P-values from this step to correct for multiple testing hypotheses, we performed a post-hoc t-test comparing the UI values at each of these hairpin sizes. The t-test compared the A3A, A3B-CTD, or A3B-full against the EV values. Once again, the P-values were corrected for multiple testing hypotheses. The bars with high statistical significance are marked with “*” ($P \leq 0.05$) or “***” ($P \leq 0.01$).

Minor comments

- Considering the similarity in the approach (genome-wide analysis of APOBEC3A/B targeting in a non-mammalian system), it might be useful to discuss the findings also in light of Taylor et al. (2013, 10.7554/eLife.00534)

Response: We have modified a sentence in the Introduction to describe these results. The new sentence reads:(Lines- 88-92) "One study showed that A3A and A3B cause mutational clusters in the yeast genome that are characteristic of breast cancer mutations (6) and another study showed that C:G to T:A mutations in a reporter gene in yeast caused by A3A preferentially occur in YTCA sites, whereas A3B favors RTCA sites (Y is a pyrimidine and R is a purine; (7))."

We also note that the preference of cytosine deaminases for TSS and tRNA genes in yeast, *E. coli* and human tumors has been described previously. The last sentence in the first paragraph of the Results accordingly now reads: (Lines 280-282) "The tendency of AID/APOBEC enzymes of preferentially deaminating cytosines near TSS and within tRNA genes in *E. coli*, yeast and human tumors has been noted previously (5,8-10)."

- Figure 4A. While adequate for the specialist, the deamination assays would benefit from a better representation (e.g. a schematic representation of the assay; the activity shown close to the labels can be confounding: the “-” sign is a label in the upper row and a value in the lower one)

Response: The enzymatic assay for cytosine deamination is shown in Supplementary Fig. S7. Additionally, a sentence has been added to the Fig. 4 legend saying "In parts A and B, the symbol “-” in rows labeled A3B-CTD indicate absence of the enzyme in the reaction, while symbol “+” represents presence of the enzyme in the reaction." The "-" signs in the row labeled "Percent Activity" are now replaced with "0" (zero).

Reviewer #3 (Remarks to the Author):

In this manuscript the authors investigated APOBEC3B (A3B) substrate preference, both in terms of 2° structure and in terms of sequence specificity. Authors extended the current knowledge base for APOBEC3B biochemical substrate specificity by showing that it displays a strong preference for hairpin-forming (vs linear) ssDNA in an UNG-deficient strain of *E. coli* (uracilation assays). This preference was also observed in human tumors by examining sequence context of APOBEC mutations and by differentiating A3B- from APOBEC3A (A3A)-derived mutations using previously reported preferences for deaminating cytidines preceded by RT and YT respectively (where Y is a pyrimidine and R is a purine). Authors were able to further differentiate A3B substrate specificity from that of A3A in *E. coli* using in vitro deaminase activity assays where A3B exhibited a preference for larger loops of 4 and 5nt, with the target cytidine in the most 3' position (A3A instead preferring C in the penultimate position of 5nt loops, and generally preferring smaller loop sizes, i.e. 3 nt as has been previously reported) and showed that, taking loop size and cytidine positioning into account, this preference is mediated by hairpin stability as calculated from stem length and composition. It is noteworthy that the authors reporting of A3B's preference for hairpins contradicts a previous report that indicated it preferred linear. The authors cite

differences in hairpin loop composition and the potential for unrecognized biases in previous studies as potential reasons for this contradiction.

The authors also examined uracilation in *E. coli* expressing full length A3B or just the C-terminal domain (compared to previous uracilation data from A3A expression), observing that the N-terminal domain of A3B has little influence on A3B deaminase activity or substrate specificity, with both full-length and truncated A3B showing a comparable preference tRNA genes and ssDNA generated on the lagging strand during replication. The authors noted increased and broader activity for full-length A3B in *E. coli*, suggesting that substrate specificity is largely conferred by A3B's C-terminal domain, and they posited that the N-terminal domain increases overall activity through increased enzyme stability or by increasing the rate at which A3B can access substrate (no data provided but consistent with prior reports).

This manuscript advances the field of APOBEC cancer biology by further characterizing A3B substrate specificity and provides a method for better distinguishing mutations caused by off-target activity from A3B versus A3A. The methodologies used and experiments were adequately rigorous, with reasoned conclusions being made. However, beyond the finding that A3B in fact does prefer hairpins the results are an incremental advance and may be more suitable for a specialized journal focused on nucleic acid modifying enzymes. The manuscript is generally well written. My specific comments follow:

1. A3B-CTD activity and full length A3B have very similar sequence specificities, but the authors indicate that full length A3B is more active in *E. coli* (lines 365-367). A western showing comparable levels of expression of the two enzymes needs to be provided to support this claim as differences in protein expression could account for this.
2. The authors identify deamination sequence preference for both A3A and A3B as a G at the +1 position and indicate this is different from previous results in yeast (Lines: 403-412). However, the results in yeast that are cited indicate preference for both enzymes of G or A at the +1 position (<https://www.nature.com/articles/ng.3378>; Figure 1), which is pretty consistent with the current data. The usage of TCA as "preference" in cited manuscript is for defining the differential Y or R nucleotide in the -2 position for A3A or A3B, respectively. The authors of the cited paper most strongly see the -2 nucleotide preference in the context of a TCA containing tetranucleotide. This section should be clarified.
3. The authors suggest the reason that A3B was previously indicated to prefer linear over hairpin sequences was due to the choice of hairpin loop sequence (lines 380-383). They indicate this is supported by poorer activity of A3B on GTTC loops compared to other loop sequences. The authors should compare the GTTC loop to a GTTC linear sequence to make this claim. The data in figure 4A actually indicates that GTTC loops are better substrates than TTC linear substrates for A3B which would indicate this is not the correct reason for the difference with the previous report (<https://www.science.org/doi/10.1126/science.aaw2872>)
4. While the authors see evidence of A3B activity at hairpins in human tumors, this activity is less obvious than for A3A and no case is made for a larger utility of this information beyond understanding the enzymology of A3B. Is there any evidence of A3B hairpin specificity being important for driver mutations, recurrent mutations, or could this information be used to better distinguish tumors experiencing A3B mutations versus A3A mutations?

Response to general comments:

We wish to thank the reviewer for concluding that our manuscript is well written and rigorous, and that it advances the field of cancer biology. However, we respectfully disagree with the reviewer that it may be more suited to a specialized journal dedicated to nucleic acid

enzymes. As the reviewer has pointed out, "...provides a method for better distinguishing mutations caused by off-target activity from A3B versus A3A." This is the broader implication of our results that would not be fully appreciated by a specialty journal. By publishing this manuscript in *Nature Communications* we hope to capture a wider audience of biologists including cancer biologists and geneticists.

Response to specific comments:

1. A3B-CTD activity and full length A3B have very similar sequence specificities, but the authors indicate that full length A3B is more active in *E. coli* (lines 365-367). A western showing comparable levels of expression of the two enzymes needs to be provided to support this claim as differences in protein expression could account for this.

Response: We thank the reviewer for their insightful suggestion. We performed the suggested experiments and the results change our interpretation of the UPD-seq data. When we performed Western blot analysis of cell-free extracts expressing A3B-CTD or A3B-full, we found unexpectedly that the full-length protein is present at much higher amounts in cell-free extracts. These data are now included in Supplementary Fig. S10 and the relevant text in the Discussion has been modified to read- (Lines 444-451) "**The only consistent difference between A3B-CTD and A3B-full was that A3B-full expression resulted in a higher number of uracilation peaks (Fig. 1A), and when comparing the same loop sequences it displayed higher UI values (Supplementary Table S3). It seemed possible that this difference between the two proteins could be due to higher expression of the latter protein in *E. coli* or intrinsically higher activity for A3B-full. To distinguish between these possibilities we performed Western blot analysis of *E. coli* cell extracts expressing the two proteins. Unexpectedly, we found that A3B-full is expressed at four- to eight-fold higher levels (Supplementary Fig. S10 and data not shown).**"

2. The authors identify deamination sequence preference for both A3A and A3B as a G at the +1 position and indicate this is different from previous results in yeast (Lines: 403-412). However, the results in yeast that are cited indicate preference for both enzymes of G or A at the +1 position (<https://www.nature.com/articles/ng.3378>; Figure 1), which is pretty consistent with the current data. The usage of TCA as "preference" in cited manuscript is for defining the differential Y or R nucleotide in the -2 position for A3A or A3B, respectively. The authors of the cited paper most strongly see the -2 nucleotide preference in the context of a TCA containing tetranucleotide. This section should be clarified.

Response: We have modified the relevant text as follows- (Lines- 485-490) "**Neither enzyme showed a strong preference for a specific base at the +1 position when all cytosines were considered (Fig. 3A) and this is consistent with the conclusions of a yeast mutational study (7). However, when the yeast mutations were examined in the TC context, YTCA was the preferred sequence for A3A, while RTCA was preferred by A3B (7). There was also a modest preference for G in the +1 position when mutations were restricted to the TC context (7), and this is not seen in our data (Fig. 3A, not shown).**"

3. The authors suggest the reason that A3B was previously indicated to prefer linear over hairpin sequences was due to the choice of hairpin loop sequence (lines 380-383). They indicate this is supported by poorer activity of A3B on GTTC loops compared to other loop sequences. The authors should compare the GTTC loop to a GTTC linear sequence to make this claim. The data in figure 4A actually indicates that GTTC loops are better substrates than TTC linear substrates for A3B which would indicate this is not the correct reason for the difference with the previous report (<https://www.science.org/doi/10.1126/science.aaw2872>)

Response: We did, in fact, compare GTT.C. loop to GTTC linear sequence. However, in Table S4 (previously Table S5) the "G" preceding the "TTC" in the linear DNA was not colored red. This gave the impression to the reviewer that the linear DNA lacked a GTTC sequence. This has now been corrected in the Supplementary Table S4.

4. While the authors see evidence of A3B activity at hairpins in human tumors, this activity is less obvious than for A3A and no case is made for a larger utility of this information beyond understanding the enzymology of A3B. Is there any evidence of A3B hairpin specificity being important for driver mutations, recurrent mutations, or could this information be used to better distinguish tumors experiencing A3B mutations versus A3A mutations?

Response: Yes, we (the Lawrence group) is using this finding to delineate the contribution of A3B to cancer genome mutations. This is presented in greater detail in the accompanying manuscript by Sanchez *et al.* from the Buisson and Lawrence groups.

In the Discussion section we discuss in greater depth why A3B may not be as strong a mutator as A3A. We point out that both A3A and A3B act predominantly on cytosines in ssDNA within replication forks, and hence RPA acts to inhibit the deamination activity of both A3A and A3B. We further speculate that the RPA may interact with the NTD of A3B to prevent its deamination activity. Sequences analogous to A3B-NTB are missing from A3A, making it less susceptible to inhibition by RPA.

Literature Cited

1. Buisson, R., Langenbucher, A., Bowen, D., Kwan, E.E., Benes, C.H., Zou, L. and Lawrence, M.S. (2019) Passenger hotspot mutations in cancer driven by APOBEC3A and mesoscale genomic features. *Science*, **364**.
2. Langenbucher, A., Bowen, D., Sakhtemani, R., Bournique, E., Wise, J.F., Zou, L., Bhagwat, A.S., Buisson, R. and Lawrence, M.S. (2021) An extended APOBEC3A mutation signature in cancer. *Nat Commun*, **12**, 1602.
3. Siriwardena, S.U., Chen, K. and Bhagwat, A.S. (2016) Functions and Malfunctions of Mammalian DNA-Cytosine Deaminases. *Chemical Reviews*, **116**, 12688-12710.
4. Hess, J.M., Bernardis, A., Kim, J., Miller, M., Taylor-Weiner, A., Haradhvala, N.J., Lawrence, M.S. and Getz, G. (2019) Passenger Hotspot Mutations in Cancer. *Cancer Cell*, **36**, 288-301.e214.
5. Sakhtemani, R., Senevirathne, V., Stewart, J., Perera, M.L.W., Pique-Regi, R., Lawrence, M.S. and Bhagwat, A.S. (2019) Genome-wide mapping of regions preferentially targeted by the human DNA-cytosine deaminase APOBEC3A using uracil-DNA pulldown and sequencing. *J Biol Chem*, **294**, 15037-15051.
6. Taylor, B.J.M., Nik-Zainal, S., Wu, Y.L., Stebbings, L.A., Raine, K., Campbell, P.J., Rada, C., Stratton, M.R. and Neuberger, M.S. (2013) DNA deaminases induce break-associated mutation showers with implication of APOBEC3B and 3A in breast cancer kataegis. *eLife*, **2**, e00534.
7. Chan, K., Roberts, S.A., Klimczak, L.J., Sterling, J.F., Saini, N., Malc, E.P., Kim, J., Kwiatkowski, D.J., Fargo, D.C., Mieczkowski, P.A. *et al.* (2015) An APOBEC3A hypermutation signature is distinguishable from the signature of background mutagenesis by APOBEC3B in human cancers. *Nat Genet*, **47**, 1067-1072.
8. Saini, N., Roberts, S.A., Sterling, J.F., Malc, E.P., Mieczkowski, P.A. and Gordenin, D.A. (2017) APOBEC3B cytidine deaminase targets the non-transcribed strand of tRNA genes in yeast. *DNA Repair (Amst)*, **53**, 4-14.
9. Sakhtemani, R., Perera, M.L.W., Hubschmann, D., Siebert, R., Lawrence, M.S. and Bhagwat, A.S. (2022) Human activation-induced deaminase lacks strong replicative strand bias or preference for cytosines in hairpin loops. *Nucleic Acids Res*, **50**, 5145-5157.
10. Taylor, B.J.M., Wu, Y.L. and Rada, C. (2014) Active RNAP pre-initiation sites are highly mutated by cytidine deaminases in yeast, with AID targeting small RNA genes. *eLife*, **3**, e03553.

REVIEWER COMMENTS

Reviewer #1 (Remarks to the Author):

In a revised manuscript, Butt and colleagues present the uracilome of A3B-CTD and A3B-full in *e. coli* as a means to define how the enzyme interacts with hairpin loops in the genome, and to discern differences of the A3B signature relative to the A3A signature. The UPD-seq data show that A3B-CTD and A3B-full act at similar regions in the genome, although in this revision the authors demonstrate a substantial overexpression of A3B-full relative to A3B-CTD in Supplemental Figure 10 which likely explains the increased activity observed with the full length enzyme. UPD-seq was used to show that hairpin loops are mutated by A3A and both A3B enzymes. In combination with biochemical assays, the authors demonstrate the specific preferences for A3A and A3B with regards to sequence context, size, and cytosine position within stem loops. These sequence/structure preferences are then evaluated in human tumor genomes.

The authors have added analyses and provided substantial clarification throughout the manuscript, which is now logical and well-written. In particular, several schematics (ie- new supplemental Figure 2) provide clarity and are helpful additions. I appreciate the authors' findings and their experimental data are robust, although I remain skeptical that the knowledge of hairpin preference by APOBECs will substantially advance our understanding of cancer pathogenesis or innate immunity. I defer to the editors as to whether the scope and impact of this work is appropriate for the journal.

Remaining queries:

Why do the authors think that the deamination activity of A3B-CTD on the GTTC hairpin substrate differs between panels 4A and 4B (50% v. 16% as I understand it)? Is this the margin of error of the biochemical assay?

Regarding overexpression of A3B-full relative to A3B-CTD. It is possible that this is dictated by the enzyme N-terminus providing protein stability as mentioned in the discussion. However it is also possible that a technical aspect of A3B expression in bacteria led to lower levels of A3B-CTD detected in extracts.

In Fig 5, the data indicate that human tumor genomes only subtly recapitulate the A3B activity as defined in *e. coli*. In particular, in Fig 5B, human cancers with C mutations in the A3B sequence context (RTCA>YTCA) there are not an enrichment of mutations in 4nt loops. It seems to me that the UPD-seq in *e. coli* and in vitro biochemical data demonstrated by the authors may be detectable in human tumors (Fig 5D, peak for A3B-most at 4nt loop), but it is very subtle and I am uncertain as to how this helps us to understand A3B activity in cancer.

Reviewer #2 (Remarks to the Author):

The Authors have satisfactorily replied to all comments from the reviewers. The manuscript is much better structured now, and some of the points raised by the reviewers have allowed the Authors to improve their interpretation.

I am convinced that passing some of the graphical explanations to the main figures would help the casual reader better understand the analyses.

In a certain sense I agree with the other reviewers that the manuscript is not groundbreaking, as it is not the companion manuscript.

On the other hand, the complementarity of the two manuscripts provides a clarification to the mode of action of the APOBEC3A and APOBEC3B that is rarely seen in other papers, usually too focused in a bidding war over which one is the baddest enzyme.

What I also like in this manuscript is its straightforwardness. It shows that a few to-the-point experiments (albeit preceded by a long technical development) can get you quite far, from an artificial bacterial assay to understanding the mutagenic processes in tumors.

Reviewer #3 (Remarks to the Author):

The authors have addressed most of my prior comments.

However, their response to my request to compare GTTC linear and hairpin substrates missed the point of the comment. The data in figure 4A shows that A3B is more active on a GTTC hairpin than TTC linear substrate. The response provided was that the TTC linear substrate is actually a GTTC linear substrate, which means that A3B is more active on the hairpin than linear for the GTTC sequence so poor activity on GTTC hairpins cannot be the reason that the previous report indicated that A3B did not preferentially deaminate hairpin sequences. A better explanation for this difference is needed. The clarification on the linear TTC substrate sequence is helpful (although it would be useful to also rename the substrate to GTTC linear in the figure).

The authors refer to a companion manuscript as a response to the lack of data indicating that A3B hairpin preference could help distinguish between tumors mutated by A3A or A3B. The overall impact of this manuscript relies on how much of an improvement the hairpin sequence specifies are in differentiating between the two enzymes. Including some analyses of this type in the current manuscript would help increase its impact.

Response to Reviewer Comments

We wish to thank once again all the reviewers for their thoughtful comments and suggestions for the improvement of the manuscript. We have made great efforts to improve the manuscript based on these comments and the re-revised manuscript is attached.

Reviewer #1

Reviewer #1 (Remarks to the Author):

In a revised manuscript, Butt and colleagues present the uracilome of A3B-CTD and A3B-full in *e. coli* as a means to define how the enzyme interacts with hairpin loops in the genome, and to discern differences of the A3B signature relative to the A3A signature. The UPD-seq data show that A3B-CTD and A3B-full act at similar regions in the genome, although in this revision the authors demonstrate a substantial overexpression of A3B-full relative to A3B-CTD in Supplemental Figure 10 which likely explains the increased activity observed with the full length enzyme. UPD-seq was used to show that hairpin loops are mutated by A3A and both A3B enzymes. In combination with biochemical assays, the authors demonstrate the specific preferences for A3A and A3B with regards to sequence context, size, and cytosine position within stem loops. These sequence/structure preferences are then evaluated in human tumor genomes.

The authors have added analyses and provided substantial clarification throughout the manuscript, which is now logical and well-written. In particular, several schematics (ie- new supplemental Figure 2) provide clarity and are helpful additions. I appreciate the authors' findings and their experimental data are robust, although I remain skeptical that the knowledge of hairpin preference by APOBECs will substantially advance our understanding of cancer pathogenesis or innate immunity. I defer to the editors as to whether the scope and impact of this work is appropriate for the journal.

Remaining queries:

Why do the authors think that the deamination activity of A3B-CTD on the GTTC hairpin substrate differs between panels 4A and 4B (50% v. 16% as I understand it)? Is this the margin of error of the biochemical assay?

Regarding overexpression of A3B-full relative to A3B-CTD. It is possible that this is dictated by the enzyme N-terminus providing protein stability as mentioned in the discussion. However it is also possible that a technical aspect of A3B expression in bacteria led to lower levels of A3B-CTD detected in extracts.

In Fig 5, the data indicate that human tumor genomes only subtly recapitulate the A3B activity as defined in *e. coli*. In particular, in Fig 5B, human cancers with C mutations in the A3B sequence context (RTCA>YTCA) there are not an enrichment of mutations in 4nt loops. It seems

to me that the UPD-seq in *e. coli* and in vitro biochemical data demonstrated by the authors may be detectable in human tumors (Fig 5D, peak for A3B-most at 4nt loop), but it is very subtle and I am uncertain as to how this helps us to understand A3B activity in cancer.

Author response

Response to general comments:

1. "... although I remain skeptical that the knowledge of hairpin preference by APOBECs will substantially advance our understanding of cancer pathogenesis or innate immunity."

Response: To demonstrate usefulness of the data resulting from UPD-seq in *E. coli*, we performed a guided non-negative matrix factorization (NMF) similar to the mutational signature analysis to evaluate the contribution of A3A and A3B in an orthogonal manner to the - 2 position preference (RTCA/YTCA). We counted the number of deaminated sites among the UPD-seq samples at sequences with different potential hairpins grouped by size of the loop and the position of the targeted C within the loop, as well as the hairpin stem strength. The combined number of deaminated sites in four A3A and six A3B-full samples were used to define the two hairpin mutational signatures, HS1 and HS2.

We then created a matrix of mutation counts from patient tumors. The contribution of the two Hairpin Signatures were estimated for each sample and the log2 ratio of the two signatures was used to classify the mutational history of samples as A3A-driven or A3B-driven. When applied to human tumor mutations they nicely separate tumors into those with A3A-like character, A3B-like character and those that have a mixed character (new Fig. 5C). This helps us understand the mutational history of these tumors and illustrates the role played by the two APOBEC enzymes in the process. The new metric was able to correctly identify all A3A- and A3B-most of the patient tumors (selected using a fraction of APOBEC mutational signature and the RTCA/YTCA character) without any information about the nucleotide context of the mutations and only using the location of the mutation with respect to their potential hairpin structures. This validates the method.

This metric was then applied to available mutational data when A3A and A3B are expressed in mice¹ and plotted against RTCA/YTCA character of the samples (X-axis). The result is shown below. In this plot, according to the RTCA/YTCA, the A3B-caused mutations should be to the left of the vertical line at 0.0 on X-axis, while mutations caused by A3A should lie to the right of that line. It is easy to see that this is not the case. In particular, many mutations in mice expressing A3B lie to the right of that line.

In contrast, according to the HS1/HS2 criterion, the A3A-caused mutations should lie above the horizontal line at 0.0, while the A3B-caused mutations should lie below that line. One can see that this is overwhelmingly the case. In other words, the HS1/HS2 method for classifying A3A/A3B mutations is superior to the RTCA/YTCA method. We have not included this analysis in our manuscript because the two murine

datasets are not strictly comparable- one is from WES, while the other is from WGS. Furthermore, the total number of samples is small compared to human tumor samples. Finally, we also did not wish to be distracted by the murine data, which involves heterologous expression of A3A and A3B, just as we have done in *E. coli* in this manuscript and others have previously done in yeast.²

Response to specific comments:

1. Why do the authors think that the deamination activity of A3B-CTD on the GTTC hairpin substrate differs between panels 4A and 4B (50% v. 16% as I understand it)? Is this the margin of error of the biochemical assay?

Response: We thank the reviewer for pointing out an inadvertent omission in the description of this experiment. The incubation time for the samples separated on the gel shown in part A of old Fig. 4 (now Fig. 4B, see below) was 20 minutes, while the incubation time for samples in part B of old Fig. 4 (now Fig. 4C) was 10 minutes. This largely explains the higher percentage conversion of cytosines in the former panel (TTC loop- 64% for former vs. 27% for latter and GTTC loop- 50% for former vs. 16% for latter. The incubation times are now noted in the figure legends.

2. Regarding overexpression of A3B-full relative to A3B-CTD. It is possible that this is dictated by the enzyme N-terminus providing protein stability as mentioned in the discussion. However it is also possible that a technical aspect of A3B expression in bacteria led to lower levels of A3B-CTD detected in extracts.

Response: It is unclear what the reviewer means by "technical aspect of A3B expression". The same strain of *E. coli* was used for the UPD-seq experiments as the Western blots and the cells were induced with the same batch of anhydrotetracycline for the same length of time (5 to 6 hours). The monoclonal antibody used for the experiment was raised against a polypeptide near the carboxy terminus of A3B and hence is found in both A3B-full and A3B-CTD. Furthermore, the presence of the NTD domain in A3B-full should have little effect on the interaction of this protein with the antibody. The experiment was repeated once and both times the result was that A3B-full showed higher expression level. The blot for the second experiment is shown below.

3. In Fig 5, the data indicate that human tumor genomes only subtly recapitulate the A3B activity as defined in *e. coli*. In particular, in Fig 5B, human cancers with C mutations in the A3B sequence context (RTCA>YTCA) there are not an enrichment of mutations in 4nt loops. It seems to me that the UPD-seq in *e. coli* and in vitro biochemical data demonstrated by the authors may be detectable in human tumors (Fig 5D, peak for A3B-most at 4nt loop), but it is very subtle and I am uncertain as to how this helps us to understand A3B activity in cancer.

Response: The A3B-most group of tumors were defined by the existence of mutations in the RTCA context in far excess over those in the YTCA context. What our data show is that the mutations in these tumors largely occur in genomic regions that are not prone to forming hairpin loops preferred by A3B. They occur mostly in extended DNA chains. This raises the question as to why do hairpins preferred by A3B in *E. coli* genome and in biochemical assays escape being targeted by A3B? We suggest that a protein factor such as RPA may interfere with the ability of A3B to target hairpin loops in the replication fork. This is the biological significance of our results and is described in separate section in Discussion (lines 533-560).

Reviewer #2 (Remarks to the Author):

The Authors have satisfactorily replied to all comments from the reviewers. The manuscript is much better structured now, and some of the points raised by the reviewers have allowed the Authors to improve their interpretation.

I am convinced that passing some of the graphical explanations to the main figures would help the casual reader better understand the analyses.

In a certain sense I agree with the other reviewers that the manuscript is not groundbreaking, as it is not the companion manuscript.

On the other hand, the complementarity of the two manuscripts provides a clarification to the mode of action of the APOBEC3A and APOBEC3B that is rarely seen in other papers, usually too focused in a bidding war over which one is the baddest enzyme.

What I also like in this manuscript is its straightforwardness. It shows that a few to-the-point experiments (albeit preceded by a long technical development) can get you quite far, from an artificial bacterial assay to understanding the mutagenic processes in tumors.

Author response

Response to general comment:

"I am convinced that passing some of the graphical explanations to the main figures would help the casual reader better understand the analyses."

Response: We have now included a schematic explanation for the biochemical assay as part A of Fig. 4.

Reviewer #3 (Remarks to the Author):

The authors have addressed most of my prior comments.

However, their response to my request to compare GTTC linear and hairpin substrates missed the point of the comment. The data in figure 4A shows that A3B is more active on a GTTC hairpin than TTC linear substrate. The response provided was that the TTC linear substrate is actually a GTTC linear substrate, which means that A3B is more active on the hairpin than linear for the GTTC sequence so poor activity on GTTC hairpins cannot be the reason that the previous report indicated that A3B did not preferentially deaminate hairpin sequences. A better explanation for this difference is needed. The clarification on the linear TTC substrate sequence is helpful (although it would be useful to also rename the substrate to GTTC linear in the figure).

The authors refer to a companion manuscript as a response to the lack of data indicating that A3B hairpin preference could help distinguish between tumors mutated by A3A or A3B. The overall impact of this manuscript relies on how much of an improvement the hairpin sequence specifies are in differentiating between the two enzymes. Including some analyses of this type in the current manuscript would help increase its impact.

Author response

Response to general comments:

1. "However, their response to my request to compare GTTC linear and hairpin substrates missed the point of the comment. The data in figure 4A shows that A3B is more active on a GTTC hairpin than TTC linear substrate. The response provided was that the TTC linear substrate is actually a GTTC linear substrate, which means that A3B is more active on the hairpin than linear for the GTTC sequence so poor activity on GTTC hairpins cannot be the reason that the previous report indicated that A3B did not preferentially deaminate hairpin sequences. A better explanation for this difference is needed. The clarification on the linear TTC substrate sequence is helpful (although it would be useful to also rename the substrate to GTTC linear in the figure)."

Response: We do not know why the 2019 paper by Buisson *et al*³ found that the GTT.C. hairpin loop was a slightly worse substrate for A3B than its linear counterpart. In their recent paper⁴, Dr. Buisson's group found that the opposite was true. See Fig. 1E from that paper reproduced below-

[redacted]

Our data in Fig. 4 is consistent with these results. To highlight this point we have added the following sentences to our manuscript- "However, in contrast with the previous report³ we found that the hairpin with GTT.C. sequence was a better substrate for A3B-CTD than its linear counterpart (Fig. 4C). The reasons for the difference with the Buisson *et al* paper³ are unclear, but we note that in a more recent publication from the Buisson group⁴, the authors found that the same hairpin was indeed a better substrate for A3B than its linear counterpart (Fig. 1E in that paper). Additionally, hairpins with TT.C. and AGTT.C. sequences were much better substrates than the GTT.C. linear (Fig. 4C). These results confirm the finding of UPD-seq experiments that A3B-CTD prefers hairpin loop substrate over an extended chain."

We have also rearranged the text in that section to make it flow better.

2. "... it would be useful to also rename the substrate to GTTC linear in the figure..."

Response: We have made the requested change.

3. "The authors refer to a companion manuscript as a response to the lack of data indicating that A3B hairpin preference could help distinguish between tumors mutated by A3A or A3B. The overall impact of this manuscript relies on how much of an improvement the hairpin sequence specifies are in differentiating between the two enzymes. Including some analyses of this type in the current manuscript would help increase its impact."

Response: We thank the reviewer for the suggestion to create a new method to distinguish between mutations caused by A3A and A3B using their hairpin loop specificities. We have indeed generated such a metric using non-negative matrix factorization (NMF) method. The result is a method to evaluate relative contributions of A3A and A3B to tumor mutations that is orthogonal to the RTCA/YTCA method of Chan *et al.*² Please see response to General Comment #1 by Reviewer #1 above.

Literature Cited

1. Durfee, C. *et al.* Human APOBEC3B promotes tumor heterogeneity in vivo including signature mutations and metastases. *bioRxiv*, doi:10.1101/2023.02.24.529970 (2023).
2. Chan, K. *et al.* An APOBEC3A hypermutation signature is distinguishable from the signature of background mutagenesis by APOBEC3B in human cancers. *Nat Genet* **47**, 1067-1072, doi:10.1038/ng.3378 (2015).
3. Buisson, R. *et al.* Passenger hotspot mutations in cancer driven by APOBEC3A and mesoscale genomic features. *Science* **364**, doi:10.1126/science.aaw2872 (2019).
4. Sanchez, A. *et al.* Mesoscale DNA Features Impact APOBEC3A and APOBEC3B Deaminase Activity and Shape Tumor Mutational Landscapes. *Nature Communications* **Submitted** (2023).

REVIEWERS' COMMENTS

Reviewer #3 (Remarks to the Author):

The additions provided by the authors in the last revision nicely addressed by previous comments. I have no further concerns.

Reviewer #3 (Remarks on code availability):

The README file only contains the statement 'Code behind APOBEC3 UPD-Seq manuscript (BioRxiv: DOI: 10.1101/2023.08.01.551518).' No instructions on installation or running is provided.

Response to Reviewer's Comments

REVIEWERS' COMMENTS

Reviewer #3 (Remarks to the Author):

The additions provided by the authors in the last revision nicely addressed by previous comments. I have no further concerns.

Reviewer #3 (Remarks on code availability):

The README file only contains the statement 'Code behind APOBEC3 UPD-Seq manuscript (BioRxiv: DOI: 10.1101/2023.08.01.551518).' No instructions on installation or running is provided.

Response

The Reviewer is referring to the code for our BioRxiv preprint, not the current manuscript. The instructions for using the code are on the website (<https://github.com/rayanramin/APOBEC3B-UPDSeq>) and are reproduced below.

Step1: Download the raw sequences using the provided accession numbers.

Samples.xlsx lists all of the samples and their accession numbers.

Four APOBEC3A samples are already published

A3A_F6	SRR6924522	PRJNA448166
A3A_H1	SRR9864913	PRJNA448166
A3A_A	SRR17822878	PRJNA801888
A3A_G	SRR17822877	PRJNA801888

New samples

| 3 A3B-CTD + 3 Empty vector control samples | PRJNA1005650 |
| 6 A3B-full (U+0009) + 6 Empty vector control samples | PRJNA1005650 |

Step2: Sequence alignment on and extracting the depth of coverage and the nucleotide counts tables.

The reference sequences are available inside the BH214_genome_files directory. Follow these steps on all samples.

<ref.fa> is the reference genome (BH214V.fa)
<plasmid_seq.fa> is the plasmid sequence (A3B_plasmids.fa)
change the to match the sample names

#index fasta files

```
bwa index -p PAB <plasmid_seq.fa>
```

```
bwa index -p BH <ref.fa>
```

alignments and filtering out plasmid reads

```
bwa mem -t 4 PAB <sample.fastq1> <sample.fastq2> | \  
samtools view -b -f4 | \  
samtools sort -n -l 6 -@ 4 - -o <sample.plasmid_aligned.bam>
```

```
bedtools bamtofastq -i <sample.plasmid_aligned.bam> -fq  
<sample.NoPlasmid_R1.fq> -fq2 <sample.NoPlasmid_R2.fq>  
bwa mem -t 4 BH <sample.NoPlasmid_R1.fq> <sample.NoPlasmid_R2.fq> | \  
samtools view -b | \  
samtools sort -l 6 -@ 4 - -o <sample.BH214.bam>
```

extracting nucleotide readcounts

```
bam-readcount -w0 -f <ref.fa> <sample.BH214.bam> | \  
awk -F ":\t|" 'BEGIN {OFS = "\t"}; {print $1, $2, $3 , $4, $21 , $35, $49 ,  
$63}' > <sample_readcount_out.txt>
```

for NDC2 analysis, the depth of coverage is needed:

```
samtools depth -aa -m 100000 <sample.BH214.bam> ><sample.BH214.depth>
```

Step3: Follow the instructions in Analysis.R to reproduce the analysis and create the Plots.

Hairpin Signature Analysis

An example MATLAB code (Hairpin_analysis.m), a test mutation set (Test_Mutations.txt), and the dependency function are available inside the Hairpin_Signature_Analysis/ directory.

Hairpin Signature Analysis is an nmf-based model to estimate the mutational activity of APOBEC3A and APOBEC3B from mutation patterns at hairpin-forming sequences. To Run the analysis on a sample's list of mutations, we first need to find the hairpin information: 1- length of the hairpin loops, 2- positions of the cytosine within the loop, and 3- stem strengths are needed.

We have created a MATLAB function to extract this information from the reference genome:

get_hairpin_info(M,ref_fasta)

This function requires the reference fasta file, and a table of mutations with the following fields:

ref_fasta: '/path/to/refernece/sequence/such/as/Homo_sapiens_assembly19.fasta'

M: A struct (table) of mutations with the following fields

chr: chromosome

pos: position of the mutation

ref: reference sequence : A/C/G/T or 1/2/3/4

alt: alternative sequence : A/C/G/T or 1/2/3/4

Note: samtools must be available at the system level to be able to run the get_hairpin_info() function!

This function Runs on a loop and, for each mutation, scans the surrounding sequence to find the potential hairpin structure.

Alternatively, one can extract the needed hairpin information by scanning the entire genome for potential hairpins.

Note:

For a large number of mutations (more than ~10,000 mutations), it is likely faster to scan the entire genome for hairpins, then map the mutations to the list of genomic positions with hairpin information and extract the loop length, loop position, and stem strength values for a set of mutations.

See <https://github.com/alangenb/ApoHP> for ApoHP: APOBEC hairpins analysis tool

Run the Hairpin Signature Analysis in MATLAB

Once the hairpin information is gathered, we can run the `hairpin_signature_analysis()` function. This function calculates two coefficients, `hs1` and `hs2`, for the Hairpin Signatures HS1 (hairpin mutation pattern of APOBEC3A) and HS2 (hairpin mutation pattern of APOBEC3B).

hairpin_signature_analysis(Hairpins,only_tpc)

This function takes in (Hairpins) the list of mutations containing the hairpin information and returns the result of the hairpin signature analysis. If a "patid"/"sapid" field is present, then the result will be repeated on each patient/sample separately. otherwise, all the mutations are assumed to be from a single sample.

input

Hairpins: A struct (table) of mutation/hairpin info with the following fields:

ref: reference sequence, numeric (1/2/3/4) : A=1, C=2, G=3, T=4

alt: alternative sequence, numeric (1/2/3/4) : A=1, C=2, G=3, T=4

looplen: length of the potential hairpin loop

looppos: position of the cytosine in the potential hairpin loop

ss: stem strength of the potential hairpin loop

minus0: base immediately 5' to the cytosine, numeric (1/2/3/4) : A=1, C=2, G=3, T=4

patid: (optional) patient/sample index, must be numeric

only_TpC: true by default; if false, then it considers all of the C:G mutations, not just TpC.

output

A nested struct which contains:

- *pat*:
 - *pat_id*
 - *nC2GT*: # of C>T | C>G mutations
 - *nTC2GT*: # of C>T | C>G mutations in TpC context
 - *tc_frac*: fraction of C:G mutations in TpC context
 - *hs1*: coefficient of HS1 signature (APOBEC3A)
 - *hs2*: coefficient of HS2 signature (APOBEC3B)
 - *log2R*: log2 ratio of *hs1* and *hs2*
 - *judgment*: 'A3A-like' or 'A3B-like'
- *counts*: The count matrix (nx90) of mutations used in the directed NMF
- *HP*: information about the 90 hairpin groups used in the analysis and HS1 and HS2 values.